# Purposeful listening in challenging conditions: A study of prediction during consecutive interpreting in noise

**Rhona M. Amos** [1]*, **Robert J. Hartsuiker** [2], **Kilian G. Seeber** [1], **Martin J. Pickering** [3]

**1** Department of Interpreting, Faculty of Translation and Interpreting, University of Geneva, Geneva, Switzerland, **2** Department of Experimental Psychology, Ghent University, Ghent, Belgium, **3** Department of Psychology, School of Philosophy, Psychology and Language Sciences, University of Edinburgh, Edinburgh, Scotland

* rhona.amos@unige.ch

**Data Availability Statement:** Raw data and scripts for all analyses are available on Open Science Framework at: https://osf.io/5dfmr/.

## Abstract

Prediction is often used during language comprehension. However, studies of prediction have tended to focus on L1 listeners in quiet conditions. Thus, it is unclear how listeners predict outside the laboratory and in specific communicative settings. Here, we report two eye-tracking studies which used a visual-world paradigm to investigate whether prediction during a consecutive interpreting task differs from prediction during a listening task in L2 listeners, and whether L2 listeners are able to predict in the noisy conditions that might be associated with this communicative setting. In a first study, thirty-six Dutch-English bilinguals either just listened to, or else listened to and then consecutively interpreted, predictable sentences presented on speech-shaped sound. In a second study, another thirty-six Dutch-English bilinguals carried out the same tasks in clear speech. Our results suggest that L2 listeners predict the meaning of upcoming words in noisy conditions. However, we did not find that predictive eye movements depended on task, nor that L2 listeners predicted upcoming word form. We also did not find a difference in predictive patterns when we compared our two studies. Thus, L2 listeners predict in noisy circumstances, supporting theories which posit that prediction regularly takes place in comprehension, but we did not find evidence that a subsequent production task or noise affects semantic prediction.

## Introduction

Listeners make predictions that speed up language comprehension [for a review, see 1]. Such predictions have been shown to include the meaning [2], the syntax [3] and the form [4] of upcoming words in a sentence. However, most theories and studies of prediction assume monolingual listeners in quiet laboratory conditions, and these conditions do not reflect many everyday communicative settings. In the following two studies, we ask whether, when challenging conditions are combined as they might be outside the laboratory, prediction takes place, which aspects of words are predicted, and whether listening purpose affects predictive processing. Specifically, we consider the case of consecutive interpreting from their second language (L2) to their first language (L1) in noisy conditions.

**Funding:** R.M.A was funded by a Swiss National Science Foundation (SNF) doc.mobility fellowship (https://www.snf.ch/en), with project number P1GEP1_188165. The funders had no role in study design, data collection and analysis, decision to publish, or preparation of the manuscript.

**Competing interests:** The authors have declared that no competing interests exist.

Reliance on prediction during comprehension may be greater when it is difficult to understand the incoming speech stream [5]. One reason for this may be that listeners are more likely to use top-down strategies to resolve ambiguities in noisy than in quiet conditions [6]. Having to attend to a degraded speech stream may lead to more top-down processing [7], suggesting that listening in noise could also increase prediction. However, sometimes listening in noise might lead to less top-down processing; for instance, strong energetic masking might lead to greater reliance on bottom-up acoustic cues [8], and comprehending in an L2 may limit prediction [4, 9, but see also 10]. So do listeners predict upcoming utterances when two challenging conditions, namely listening in an L2 and listening in noise, are combined, and, if so, which aspects of utterances do listeners predict in these conditions? Does L2 prediction in noisy conditions differ from L2 prediction in quiet conditions? Understanding whether and how prediction is limited in the face of combined adverse conditions may shed light on whether prediction is costly, and thus inform our understanding of the mechanisms which underlie prediction during comprehension.

We also ask whether an interpreting task affects prediction. Prediction may take place using the production mechanism, and engaging the production mechanism in utterance planning may therefore influence predictive processing [1]. We consider prediction during consecutive interpreting, a mode of interpreting in which interpreters listen to the speaker (often in their L2), and subsequently interpret what the speaker has said (often into their L1). This task thus requires focused attention and engagement of the production mechanism. Engaging the production mechanism may support prediction [11], particularly when production and comprehension are closely aligned [12]. We thus ask whether engaging in a consecutive interpreting task increases predictive processing. Understanding whether listener aims and utterance planning affect prediction during comprehension may shed light on whether the extent of prediction depends on the communicative scenario, and whether engagement of the production mechanism supports prediction.

In this paper, we first review evidence of prediction during comprehension in L2 before considering, based on the current state of knowledge in the field, what the influence of noise and a consecutive interpreting task may be on such prediction. We then present the results of two studies. The first study considers the effect of consecutive interpreting on L2 prediction in noisy conditions, and the second (follow-up) study considers the effect of consecutive interpreting on L2 prediction in quiet conditions. We then compare the results of the studies and consider the potential influence of L2 proficiency on L2 prediction in noise.

## Prediction during comprehension in L2

There is evidence that L2 listeners predict aspects of an upcoming utterance before they have begun to hear (or read) the utterance. For instance, Dijkgraaf, Hartsuiker and Duyck [13] had a group of Dutch-English bilinguals and a group of English monolinguals listen to sentences that were either constraining, or not, for a particular noun (e.g., "Mary *knits* a scarf" or "Mary *loses* a scarf"). Dutch-English bilinguals listened in both Dutch and English, while English monolinguals listened in English only. Measurements of participants' eye movements to a screen displaying four objects, only one of which could be knitted (but all of which could be lost), showed that all groups made predictive eye movements and that the effect of condition was similar across groups. L2 readers also anticipate upcoming words and their articles during reading when presented with highly constraining sentences that are syntactically similar to sentences in their L1 [10]. Predictions made in L2 are robust enough to leave a trace in memory, with late bilinguals not only predicting upcoming words, but also subsequently identifying these words as having been heard (even when the predictable words had been muted) [14].

How might such prediction take place? There is evidence that, during conversation, people plan their own utterance while listening, meaning that they both make predictions about what they and their interlocutor will say and engage their production mechanism during comprehension [15]. For instance, corpus analyses show that gaps between interlocutors tend to be short, typically around 200 ms [16], and contributions sometimes overlap [17]. It may be that this concurrent engagement of the production mechanism during comprehension leads to prediction-by-production, in which listeners form predictions about upcoming content by means of covert imitation [1, 11]. Prediction-by-production may support prediction during comprehension because covert imitation using the production mechanism leads listeners to go through the same steps in the same order as the original speaker to predict the meaning, syntax and form of upcoming words [see 1 for a detailed theoretical account]. On this account, syntax and form would be predicted at a later stage than meaning.

However, L2 listeners' predictions may be delayed compared to those of L1 listeners [18], and they do not always appear to predict syntax and form. Martin et al. [9] found that although both L1 and L2 listeners showed an N400 effect on an unpredictable consonant or vowel-initial noun in a reading task, suggesting that both groups predicted content on a semantic level, only L1 speakers showed an N400 effect on an article that corresponded with the upcoming noun in grammatical gender. This suggests that L2 listeners did not predict the phonological form of the upcoming noun. Similarly, Ito et al. [4] found that L2 listeners did not make phonological predictions, whereas L1 listeners did. Mitsugi and Macwhinney [19] found no evidence of syntactic prediction in L2, and Koch, Bulté, Housen and Godfroid [20] found that while L2 listeners used morphological information to predict, they did so more slowly than L1 listeners. This suggests that L2 prediction is impoverished compared to L1 prediction [see also 21]. However, there is considerable diversity among L2 listeners [22], and high-proficiency L2 listeners may be able to use syntactic information, for instance, to form predictions just as L1 speakers do [23, 24]. This evidence suggests that some parts of predictive processing (e.g., semantic prediction) may take place when resources are more limited (such as in L2 comprehension) but that other parts (e.g., syntactic and phonological prediction) may require resources. Thus prediction may take place on the semantic level without taking place on the phonological level, but not the reverse, suggesting that this semantic prediction may take place earlier and/or may be less costly than phonological prediction.

The evidence thus demonstrates that prediction takes place during L2 comprehension, but that prediction in L2 may be more limited than prediction in L1. Prediction may take place using the production mechanism, in which case, we would expect semantic prediction to take place earlier than phonological prediction. However, some stages of the prediction process may be more costly; for instance, more cognitive resources may be required for phonological prediction than semantic prediction.

## Listening to speech in noise

Listening to speech in noise is more challenging than listening to speech in quiet conditions, and overall comprehension performance decreases when listening to speech in noise [25]. The difficulty experienced by listeners, and the way in which they process speech in noise (including the extent to which they predict), may depend on factors related to both the noise, for instance the type of noise (or masking) and the signal-to-noise ratio (SNR), and the listener, for instance language of comprehension (L1 or L2), and degree of attention.

**Types and levels of noise.**  Mattys, Davis, Bradlow and Scott [26] distinguished between energetic and informational masking. Energetic masking refers to interference from another source that temporally overlaps with the target signal (e.g., grey noise). Informational masking

refers to any additional distraction caused by a competing signal over and above its energetic interference with the speech stream (for instance, when there is a competing speaker whom the listener understands). In the case of energetic masking, the lower the SNR, the more difficult speech is to understand [25]. Listeners appear to engage working memory to process energetically masked speech, but only at intermediate SNRs (e.g., -2 dB and 0 dB). At SNRs above this, speech recognition may be easy and require less top-down processing, and at SNRs below this, listeners appear to stop actively listening to the speech stream [27]. This suggests that cognitive resources are engaged to process noisy speech at intermediate SNRs in particular. Meanwhile, an informational masker interferes with speech comprehension as, in addition to any energetic masking, it leads to divided attention between two sources of information.

**Top-down language processing in noise in L2.** Listening to noisy speech in an L2 is particularly challenging [for a review, see 28]. This may be because L2 listeners are less able to use contextual information to help them understand noisy speech. For instance, Krizman, Bradlow, Lam and Kraus [29] found that L1 listeners were significantly better than L2 listeners at sentence comprehension in noise, slightly better than L2 listeners at word perception in noise (but not significantly better), and worse than L2 listeners at tone perception in noise. This shows that the L2 disadvantage in noise is specifically linked to linguistic stimuli, and that the L1 comprehension advantage is more pronounced when more contextual cues are present. Similarly, Shi [30] also found that although both L1 and L2 listeners benefitted from context effects when listening to acoustically degraded speech, L1 speakers (and simultaneous bilinguals) benefitted more than L2 speakers [see also 31, who found that context helped L1 but not L2 listeners perceive words in noise].

There is also fMRI evidence that non-native listeners do not engage in top-down processing in the same way as native speakers when listening to noisy speech. Rammell, Cheng, Pisoni, and Newman [32] considered the neural correlates of English-Spanish late bilinguals when listening to speech in either L1 or L2. When participants listened to speech in noise in their L2, brain regions linked to auditory language processing were activated. When the same participants listened to speech in noise in their L1, brain regions linked to executive functions were activated. This suggests that participants engaged in bottom-up processing of the speech signal in their L2, and top-down language processing in their L1 [see also 33].

**Prediction in noise in L2.** The evidence reviewed above suggests that L2 listeners have more difficulty listening in noise than L1 listeners, and that this difficulty may be linked to an inability to use contextual and semantic cues to aid perception of speech in noise. Since prediction during comprehension may rely on the use of context [34] and semantics [2], L2 listeners may also be unable to use such cues to form predictions during comprehension in noise. Indeed, Mayo, Florentine and Buus [35] had L1 and L2 participants listen to predictable and less predictable sentences presented at different SNRs and write down their final word. L1 listeners and L2 listeners classed as early bilinguals were able to provide the final word accurately at a significantly lower SNR than L2 listeners classed as late bilinguals, and sentence predictability did not affect the performance of this second group of L2 listeners.

In addition, L2 listening may be generally more effortful than L1 listening. For instance, semantic integration may be slower for L2 than L1 listeners [36], and L2 listeners may find it more difficult to access grammatical knowledge [37]. As previously reviewed, predictive processing in L2 may also be delayed compared to L1 [e.g., 20]. Such difficulties may be compounded by additional processing difficulty due to noise. For instance, listeners begin lexical access later in noise-vocoded than in clear speech [38], and noise lengthens the time it takes for listeners to launch saccades to target objects in a display [39]. Thus, parts of L2 processing may be slower than L1 processing, and noise may slow down this processing further, so that semantic and grammatical knowledge may not be available in time for L2 listeners to make

predictions. The joint effect of listening in an L2 and listening in noise could therefore prevent prediction in L2 from taking place.

However, predictive processing in L2 listeners in noise could be influenced by the extent to which their attention is focused on what they hear, as focused attention has been shown to improve perception of speech in noise. Wild et al. [7] had participants listen to speech that varied in acoustic clarity, and asked them to attend to either the speech, an auditory distractor, or a visual distractor. A post-scan recognition test showed a significant interaction for speech type and attention, suggesting that the recognition of moderately degraded speech was significantly enhanced by attention to the speech, whereas attention did not affect recognition of clear speech or extremely degraded speech. Clarke and Garrett [40] also found post-hoc evidence that attention may play a role in processing noisy and accented speech: Listeners who had listened to speech in noise adapted more quickly to accented speech than those who had listened to clear speech before listening to accented speech. Similarly, after training in listening to speeded words, listeners allocate working memory resources more effectively for faster recognition, potentially reducing demands on working memory [41].

Given that L2 listeners do not appear to use contextual and semantic cues as L1 listeners do when listening to speech in noise, it is unclear whether L2 listeners form predictions when listening to noisy speech. However, focused attention in noise may influence prediction in L2 in noise.

## Consecutive interpreting

Consecutive interpreting is a mode of interpreting in which an interpreter listens to a message in one language and then reproduces the same message in a different language. The interpreter waits until the speaker has finished speaking (momentarily or definitively) before producing the interpretation (unlike in simultaneous interpreting, when the interpreter begins producing the interpretation while still listening to the original speaker). Unlike simultaneous interpreters, consecutive interpreters do not typically work in sound-proofed booths, and background noise may thus be a feature of their work environment. For instance, they might work for press conferences and guided tours [42], as well as in hospitals and police stations [43], or they might work online, which may also lead to background noise [44].

The consecutive interpreter's main aim is to understand the utterance fully in order to reproduce the salient content–an aim that is very different from communicative settings in which the comprehender listens in order to contribute to dialogue. The interpreter must listen with great concentration, to the entirety of the content with the purpose of "retelling" what she has heard [45]. In other words, the interpreter does not contribute to developing (the content of) the message, but only to its formulation. In addition, the "retelling" of the utterance in another language involves memory, and the focused attention necessary to remember what one has heard [46] may be greater than in other forms of bilingual listening [47]. Listener goals and strategies have been shown to influence predictive processing [48], as have instructions to predict [49]. Thus, listening with the aim of "retelling" an utterance might lead to greater prediction. Such focused attention may also improve perception of speech in noise (as reviewed above), and so in noisy speech, the role of attention may be particularly important.

Meanwhile, models of the simultaneous interpreting process ascribe a role to prediction [50–52], and some of the potential benefits of prediction in simultaneous interpreting also apply to consecutive interpreting. For instance, prediction may allow interpreters to shift more of their focus to production, rather than comprehension [53], and thus produce their own utterances more rapidly [54].

As we have noted, some models of prediction also ascribe a role to the production mechanism in prediction [1]. During a consecutive interpreting task, listeners might both listen and prepare their own utterance (which should have the same content as the original utterance) at the same time. While listening to consecutively interpret, listeners must memorise content in order to reproduce it in the target language (which may encourage covert imitation) and may also begin to plan their own utterance while listening by making use of parallel activation of target language equivalents [55]. Such parallel activation of the target language may be particularly likely when interpreting from L2 into L1 [56], which is standard for interpreters working in Europe [57]. This activation of the production mechanism during listening may support prediction via the production mechanism, a hallmark of which could be phonological prediction [1]. Thus, listening for the purpose of consecutive interpreting may be more likely to involve phonological prediction than listening for the purpose of comprehension. Zhao, Chen, and Cai [58] found that bilinguals read predictable but not unpredictable words faster in their L2 when they were reading to consecutively interpret rather than to recall. This suggests that prediction might be enhanced during comprehension when interpreting consecutively. If predictive processing takes place earlier during a consecutive interpreting, it may also be more likely to involve prediction of form, as listeners may predict meaning before form.

Thus, L2 prediction may be limited in noise. However, focused attention may improve L2 speech perception in noise, and focused attention and engagement of the production mechanism may enhance prediction. Prediction in L2 in noise may thus be enhanced during a consecutive interpreting task compared to during a listening task–potentially taking place earlier and to a greater extent, and/or including prediction of form. Equally, in quiet conditions, a consecutive interpreting task may also lead to enhanced prediction compared to a listening task.

## The current study

We designed two studies. In a first study (hereafter: Noisy Speech study) we first asked: do proficient L2 listeners make semantic predictions during comprehension in noise? Second, is prediction enhanced during a consecutive interpreting task due to focused attention and engagement of the production mechanism: does it take place earlier, to a greater extent and/or include phonological prediction? In a follow-up study (hereafter: Clear Speech study), we asked: do predictive patterns during a listening and a consecutive task differ in clear listening conditions? We then compared the two studies, asking: does noise affect patterns of prediction in L2 listeners?

In both studies, we tested Dutch-English bilinguals, who were recruited from Ghent University's Faculty of Translation, Interpreting and Communication or from the participant pool at Ghent University's Psychology Faculty. Participants listened to a highly constraining sentence, such as "*Bob proposed and gave her a ring that had cost half his monthly wage*" and viewed a visual array containing three distractors and an image corresponding to a critical word: either the predictable word (e.g., *ring*), a word phonologically related to the English form of the predictable word (e.g., *ribbon*), or an unrelated word (e.g. *letter*).

Despite the combination of challenging listening conditions for the Noisy Speech study, we thought it likely that high-proficiency L2 listeners would fixate more on the image representing the predictable word than on the image representing the unrelated word before hearing the predictable word, in line with studies showing that L2 listeners do make predictions [e.g., 13]. However, as L2 listeners do not appear to make effective use of contextual effects to improve speech perception in noise [30, 32, 35], we also considered that they might not make predictions in noise either.

We also hypothesized that a consecutive interpreting task would lead to earlier and more predictive fixations on images representing predictable words, and may lead to predictive fixations on phonological competitors after fixations on these predictable images. Consecutive interpreting requires focused attention and this might both improve speech recognition in noise [7] and encourage earlier or greater prediction [48, 58]. In addition, consecutive interpreting engages the production mechanism, and this might lead to word-form prediction [1]. Phonological information about upcoming words may be pre-activated when listeners hear highly predictable sentences [4, 59] and this may drive looks to objects depicting phonological competitors [4], as linguistic input modulates the level of activation of different objects on a display, leading listeners to fixate on the most relevant object [60, 61]. Although previous studies [e.g., 4] did not find evidence of word form prediction in L2 learners of English, word-form prediction might take place when prediction is particularly advantageous [e.g., 48], when listening effort is maximised [e.g., 7], and when the production system is engaged in planning the same utterance in a different language; in other words, in a consecutive interpreting task in noise.

In our Clear Speech follow-up study, we hypothesised that the lack of a phonological effect for the consecutive interpreting task in our first study might be because participants could not predict as effectively in noise, as noise slows down predictive processing and word form may be predicted after meaning. Thus, in Clear speech, L2 participants might make phonological predictions when carrying out an interpreting task. We also hypothesized that patterns of prediction between Noisy and Clear speech would differ, because L2 prediction in noise may be impoverished compared to L2 prediction in quiet laboratory conditions. However, we also considered the possibility that L2 prediction in noise, particularly in the consecutive interpreting task, may in fact be enhanced by top-down processing.

## Materials and methods

### Participants

Thirty-six Dutch-English bilingual young adults with normal hearing participated in each study. We determined sample size based on [4], which used items (sentences and pictures) with similar characteristics, and an experimental structure close to that of our experiment. Each of our two studies had 50% more participants than Experiment 1 of [4] and had five items per condition per task [as compared to four per condition in 4]. Participants provided written consent by signing an informed consent form approved by the Ethics Committee of the Faculty of Psychology and Pedagogical Science at Ghent University.

Participants completed a language background questionnaire, based on the Leap-Q questionnaire [62]. Unlike in the original questionnaire, participants were asked to provide language background information only on Dutch and English (rather than on all of their languages). Self-proficiency ratings were made on a 10-point scale. Participants also stated whether they had received any training in consecutive interpreting, and how much training if applicable. They also completed a LexTALE test at www.lextale.com [63]. Proficient users of English would be expected to score between 80 and 100% [63] on this test. Based on the background information collected (Table 1), we consider that our participants were proficient in English. We could therefore be relatively confident that all of our participants would comprehend the English sentence, associate the pictures and words used and complete the consecutive interpreting task, thus ensuring conditions in which prediction could take place and the two tasks would differ.

**Exclusions.** For the Noisy Speech study, we excluded two participants because they fixated the visual stimuli less than 3% of the time in the consecutive interpreting task [following 4].

**Table 1. Background information for participants in the Noisy and Clear Speech studies.**

|  | Noisy Speech Study | Clear Speech Study |
|---|---|---|
| Age *(yrs)* | 23.47 ± 3.28 | 22 ± 2.43 |
| Languages spoken | 4.3 ± 0.92 | 4 ± 0.84 |
| Age *(yrs)* of Acquisition (Dutch) | 1.31 ± 1.26 | 1.08 ± 1.17 |
| Age *(yrs)* at which became proficient at reading Dutch | 7.1 ± 2.5 | 6.7 ± 1.9 |
| Time *(yrs)* living in Dutch-speaking area | 22.56 ± 3.27 | 21.67 ± 2.01 |
| Current exposure to Dutch % | 56.14 ± 15.87 | 63.22 ± 15.99 |
| Age *(yrs)* of Acquisition (English) | 10.61 ± 2.96 | 10.25 ± 3.56 |
| Age *(yrs)* at which became proficient at reading English | 15.03 ± 3.56 | 14.22 ± 2.73 |
| Time *(mths)* living in English-speaking area | 6.24 ± 21.79[a] | 2.70 ± 11.12[b] |
| Current exposure to English % | 19.36 ± 9.76 | 24.42 ± 13.37 |
| Self-rated speaking ability (English) | 7.81 ± 1.21 | 7.78 ± 0.99 |
| Self-rated reading ability (English) | 8.36 ± 0.93 | 8.47 ± 0.94 |
| Self-rated listening ability (English) | 8.19 ± 0.98 | 8.31 ± 1.28 |
| LexTale score % | 85.46 ± 8.80 | 86.05 ± 8.26 |
| Participants with consecutive training | 15 | 12 |
| Amount of consecutive training *(mths)* | 1–4[c] | 1–3 |

[ab]One participant reported living in an English-speaking environment for 23 years and one for 22 years in the Noisy and Clear Speech studies respectively. These were clearly reporting errors as the participants were aged 23 or 22 respectively and currently living in Belgium.

[c]Two students had received 10 months of training in consecutive interpreting (but had not yet been awarded their degree)

One participant was excluded because they did not complete the consecutive interpreting exercise more than 50% of the time. (The participant either did not say anything or else said "bla bla bla"). One additional participant was tested but excluded at random because there was a complete rotation of condition and position of the critical image for each item within each task and between tasks with 36 participants. For the Clear Speech study, we excluded one participant who fixated the visual stimuli less than 3% of the time in the consecutive interpreting task.

## Materials

Experimental stimuli consisted of 30 English sentences (see S1 Appendix), each paired with one of three visual arrays. The experimental sentences each contained a highly predictable word (e.g., *ring* in "Bob proposed and gave her a *ring* that had cost him half his monthly wage.") at varied positions in the sentence (range: 5 – 16th word) but never sentence finally. The experimental sentences consisted of a mean of 15.6 words (range: 10–21, SD: 2.79). They were read at a mean rate of 2.05 syllables per second (SD = 0.24, Range: 1.61–2.65) by a male native speaker of Southern British English in a sound-proof booth. The average sentence duration was 9.90 seconds (SD = 1.68, Range: 5.92–12.98). The onset of the critical word was, on average, at 6.26 seconds (SD: 2.00, Range: 3.04–11.3). In addition, there were 30 filler sentences of a similar length. These sentences were designed to not be constraining for any particular word after the verb, for instance "They chose the holiday destination because they wanted to see *dolphins* in the sea". Sentences were taken from Ito et al. [4] and Block and Baldwin [64], or else were designed by the authors. The intensity of all sentences was set to 70 dB using Praat

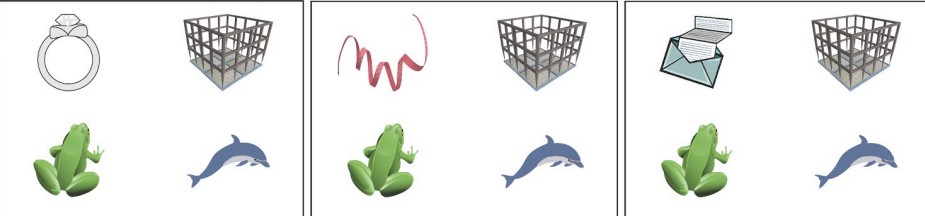

**Fig 1. Example of the three conditions of the visual scene.** Example for the sentence "Bob proposed and gave her a ring that had cost him half his monthly wage". The critical image appears in the top left-hand corner in the, from left, Target (ring), Competitor (ribbon) and Unrelated (letter) condition.

and the volume at which stimuli were presented was held constant across participants and trials in both studies.

Each of the visual arrays contained four objects: a critical object and three distractor objects (Fig 1). In the target condition, the critical object corresponded to the predictable word (e.g., *ring*). In the Competitor condition, the English name of the critical object phonologically overlapped at onset with the predictable word (e.g., *ribbon*). The mean number of phonemes shared between the predictable words and the competitor words was 2.2 (SD 0.55) out of a total of 3.6 (61%). In the unrelated condition, the onset of the name of the critical object did not overlap with the English name of the object. Using a visual array of different objects, rather than a visual scene including contextual information, allowed us to reduce any potential influence of visual context on linguistic processing [see 65 for further discussion].

The predictability of the target words was assessed by a group of undergraduate native speakers of Dutch, using an online cloze probability test. Twenty first-year undergraduate native speakers of Dutch who spoke English as a second language read the sentences truncated before a predictable word and provided that word in English. Mean cloze probability was 84.3% (SD: 17.21%, Range: 45–100%) (see Table 2). Another group of 48 L1 Dutch Bachelors and Masters-level students, who did not participate in the cloze probability test, named each image used in the experiment. Each image was rated by at least 12 students. When names provided had the same meaning and the same phonological onset, they were counted as the same word (e.g., life jacket/life vest). Two items were not included in the norming study due to an error (fish/finger). The naming agreement for images in all conditions, as well as for the distractor images, is shown in Table 2.

The study used 30 experimental and 30 filler sentences, as well as 30 matched versions of the visual stimuli, shown once with an experimental and once with a filler sentence. The stimuli were shown with a filler sentence so that we could check whether participants fixated on competitor or target objects even when they were presented with a sentence that was not constraining for a predictable word, allowing us to exclude the possibility of visual bias towards particular images. The matched visual stimuli showed the same four images, but the quadrants

**Table 2. Results of norming studies.**

| Sentence cloze rating % | Naming agreement (Target) % | Naming agreement (Competitor) % | Naming agreement (Unrelated) % | Naming agreement (Distractors) % |
|---|---|---|---|---|
| **84.3** | **90.8** | **81.6** | **84.2** | **84.3** |
| SD: 17.21 | SD: 16.7 | SD: 23.5 | SD: 21.7 | SD: 21.4 |
| Range: 45–100 | Range: 16.7–100 | Range: 8.3–100 | Range: 16.7–100 | Range: 8.3 to 100 |

in which the images appeared were varied. Filler sentences mentioned distractor objects 66.7% of the time, so that together with the experimental sentences, in which the critical object was present 33.3% of the time (i.e., in the target condition), the sentences mentioned one of the objects in the visual scene 50% of the time.

We constructed three sets of experimental items, each containing 10 items of each experimental condition (target, competitor and unrelated) and 30 filler sentences. These three sets of items were then each divided into two half lists. The matched visual stimuli were assigned to different half lists, meaning that in one half list the visual stimuli appeared with an experimental sentence, and in the other half list they appeared with a filler sentence. The half lists were then recombined in two different orders to create 6 full lists. In a few instances, an image that was a phonological competitor or target object was mentioned in another sentence and so it was always shown as a competitor or target object first. The half lists were then pseudo-randomized. Critical objects appeared in each of the three conditions in each of the four quadrants equally frequently for each set of items.

For the Noisy Speech study only, we added speech-shaped sound to the sentences at a SNR of 0 dB, with sentences and noise cut at the nearest zero crossing using Praat. The sound was created by demodulating ICRA 7 multi-speaker babble from the database at http://www.icra-audiology.org [66]. We used speech-shaped sound because it provides a constant and stationary masker [28] covering the same frequency spectrum as multi-speaker babble. This allowed us to create the same conditions in the different trials in our study. We chose an SNR of 0 dB, because at 0 dB sentences can easily be understood, but isolated words may be less intelligible [67], and young adults with normal hearing should be able to repeat sentences correctly at least 90% of the time, without relying on visual input [68]. The predictable words are likely to have been recognised more frequently than this, because of the contextual constraints [69 estimated close to 100% word recognition for young adults in high-constraint sentences with no visual context]. At SNRs below 0 dB, only some parts of the speech signal are likely to be heard over the noise [28], and at SNRs of below -2 dB, listeners may disengage from a listening task [27]. We thus created a noise level at which it would still be possible for our listeners to rely on linguistic (rather than visual) input and top-down processing, and consequently interpret the sentences.

## Procedure

**Picture familiarisation.** Each study started with a picture familiarization task. Participants saw the English names of the objects and heard their names spoken by the same speaker who recorded the experimental sentences. Participants then saw the objects a second time and provided their names in English. The mean picture naming accuracy on first attempt was 97.7% (SD: 4.9%) and 97.5% (SD: 4.8%) for the Clear and Noisy speech groups respectively. Incorrectly named objects were repeated once, and then the experimenter provided participants with the correct name.

**Eye-tracking studies (Noisy and Clear Speech).** In the eye-tracking experiment, participants were seated in front of a computer screen in the EyeLink+ laboratory at Ghent University. Participants' right eye was tracked, except when participants knew that their left eye was more dominant, an initial calibration of the right eye showed an uneven calibration grid, or reflections on glasses interfered with calibration or validation of the right eye. An SR Research Eyelink 1000 Plus desktop-mounted eye-tracker was used in the remote mode. Auditory stimuli were presented at a constant volume through a pair of Sennheiser over-ear headphones.

After participants viewed the instructions, the eye-tracker was calibrated using the nine-point calibration grid. The experiment comprised two halves, one corresponding to the

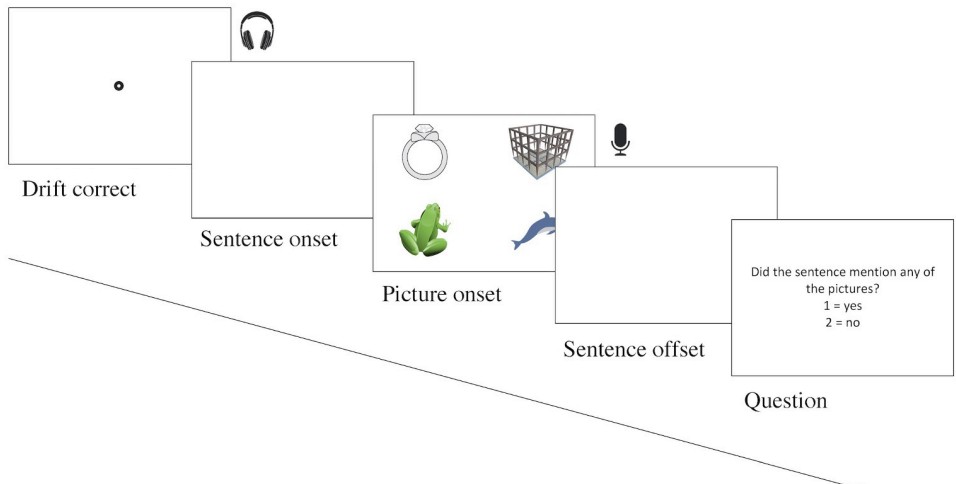

**Fig 2. Experimental procedure for an experimental trial.** From left to right: 1., Drift correct. 2., A blank screen is shown and the sentence begins. 3., 1000ms before predictable word onset the visual array is shown (here in the Target condition). 4., The sentence finishes and a blank screen is shown. In the consecutive task participants have 12000 to complete their interpretation. 5., The comprehension question appears.

listening task and the other to the consecutive task, with task order counterbalanced across participants. Each half started with two practice trials, after which participants were given a chance to ask questions, and the eye-tracker was re-calibrated before participants began each main session. Pictures were presented on a viewing monitor at a resolution of 1024 x 768 pixels. Each experimental trial took place as follows (see Fig 2 for a graphic illustration). A drift correct was performed, the sentence began, and then the visual array was presented 1000ms before onset of the predictable word in experimental trials. On filler trials, the presentation was 1000ms before the onset of a word that referred to a distractor, or else at an arbitrary mid-sentence point if the sentence did not mention anything in the array. This short preview period limited the time during which participants could form expectations based on the visual display (rather than the linguistic input), given that conceptual preparation during picture naming takes around 200ms [70]. The pictures stayed on the screen until offset of the spoken sentence. In the listening half, a blank screen was then shown for 1000ms before participants were presented with the question "Did the sentence mention any of the pictures?" In the consecutive interpreting half, a blank screen was then shown for 12000ms while participants consecutively interpreted the sentence into Dutch, and then the same question was presented. Participants answered yes or no using keys 1 or 2 respectively on their keyboard.

## Results and discussion

### Comprehension question accuracy (Noisy and Clear Speech)

For the Noisy Speech study, the mean accuracy in the comprehension questions for the experimental items was 97.9% (SD: 2.38%). Correct responses were evenly distributed between the listening and consecutive tasks, with accuracies of 98.0% (SD: 3.09%) and 97.8% (SD: 3.13%) respectively. For the Clear Speech study, the mean accuracy in the comprehension questions for the experimental items over both tasks was 99.3% (SD: 1.3%). This broke down evenly across the listening and the consecutive tasks, with response accuracies of 99.4% (SD: 1.7%) and 99.1% (SD: 2.1%) respectively. Incorrectly answered trials were excluded from the analyses

following Ito et al. (2018). In the Noisy Speech study, in the consecutive interpreting task, participants produced the exact Dutch translation of the predictable word in 89.6% (SD: 10.8%) of experimental trials. In the Clear Speech study, participants produced the exact Dutch translation of the predictable word in 92.2% (SD: 10.6%) of experimental trials. Participants began their interpretation at 1148ms (SD: 954ms) after sentence offset in the Noisy Speech study and at 1146ms (SD: 789 ms) after sentence offset in the Clear Speech study. In four of 1080 trials (0.37%) in the Noisy Speech study and two of 1080 trials (0.19%) in the Clear Speech study, no interpretation was provided. We did not exclude trials in which participants did not produce the critical word or the interpretation, as we reasoned that even when participants did not produce an exact translation, they were still concentrated and engaged in utterance planning while listening in this consecutive task. Importantly, in this way, we applied the same exclusion criteria to both halves of the experiment.

## Eye-tracking data analysis

We first analysed our data by task using a linear mixed model with the lme4 package [71] in R Studio Version 1.2.5033 [72] using the optimx optimizer. We considered the Noisy and Clear speech studies separately. Proportions of time spent fixating on target, competitor and unrelated objects were calculated separately using the EyeLink's DataViewer for 50ms bins. Blinks and fixations outside the computer screen were included in the calculation of the proportion of fixations. However, bins containing only blinks or fixations outside the computer screen were excluded from the analysis [following 4]. We explored the time-course of effects by running a first model for each bin from 1000ms before target word onset to 1000ms after onset. The model evaluated the arcsine-transformed fixation proportions on critical objects (dependent variable) as predicted by condition (independent variable) for each bin. The unrelated condition was used as a reference group (using the relevel function in RStudio) so that we could test the effects of each critical condition relative to the unrelated baseline condition. In order to check whether the order in which participants completed the tasks affected predictive fixations, we also included task order as an independent variable and checked for an interaction with condition. As in [4] and similarly to [73], we base our conclusions on the earliest time at which a minimum of three consecutive bins diverge significantly between the target or the phonological competitor condition and the baseline condition. We consider that the difference between a critical and baseline condition is significant when $|t|>2$ [74]. We also ran a second model for both tasks together, again considering the Noisy and Clear speech studies separately. Here again our dependent variable was fixation proportions on the critical object as predicted by condition and task (fixed effects). In all linear mixed models, we included random slopes and intercepts by participants and by items [75].

We then looked at both studies together, and considered 1. whether there was any difference between the Noisy and Clear speech groups and 2. whether other parameters (Consecutive Interpreting training and LexTale score) might have affected our findings.

## Results of eye-tracking studies

**Noisy Speech study—Experimental items.** We first verified whether task order interacted with condition to affect fixations on predictable objects. In the listening task, we did not find an interaction in any time bin between task order and condition. We therefore dropped the interaction term from our model, and considered whether fixations depended on condition in the listening task. Participants began looking significantly more at the target object from -450ms before word onset until at least 1000ms after onset of the predictable word (Fig 3).

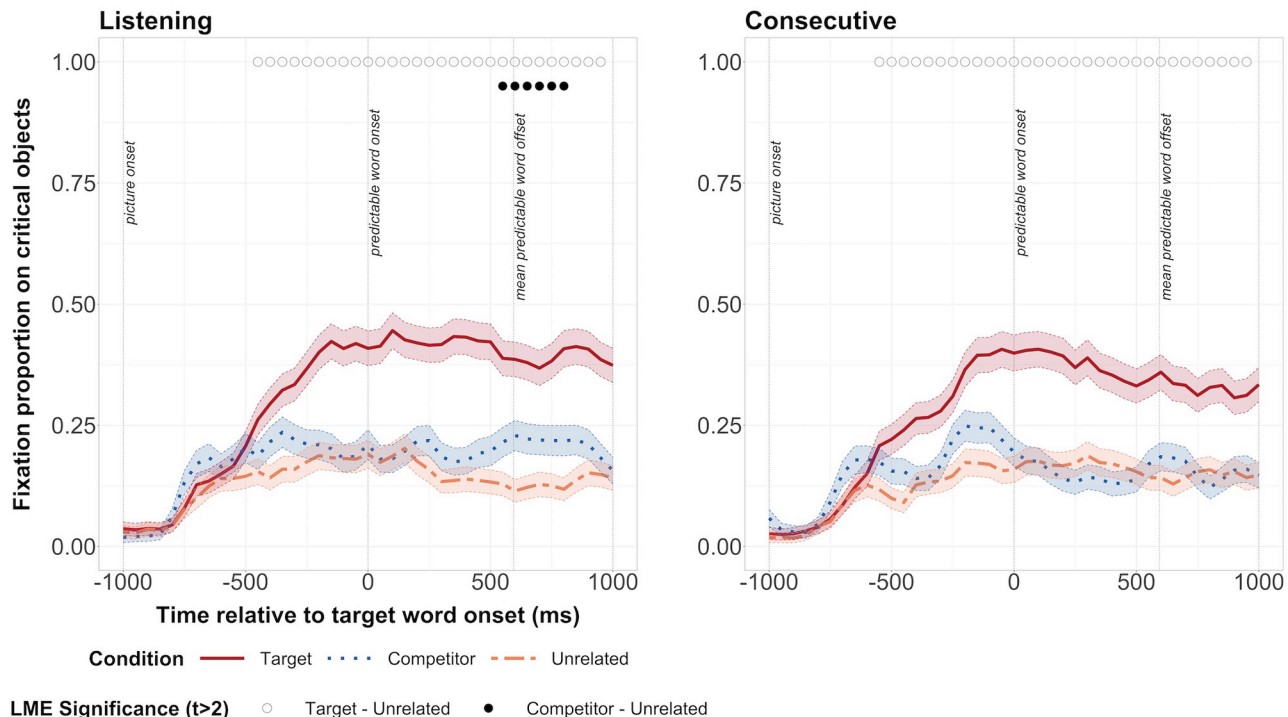

**Fig 3. Fixation proportions on target, competitor and unrelated objects in the Noisy Speech study.** The listening task is shown on the left, and the consecutive task on the right. Open circles along the top represent bins during which there was a significant difference between fixation proportions on target and unrelated objects. Filled circles represent bins during which there was a significant difference between fixations proportions on competitor and unrelated objects. Transparent thick lines are error bars representing standard errors.

Participants also looked significantly more at the competitor object compared to the target object from 550ms after word onset until 850ms after word onset.

In the consecutive task, we again did not find any interaction at any time bin between task order and condition (Fig 3). However, we did find a main effect of task order over three bins from -250ms to -100ms, suggesting that when people completed the listening task first, the difference in fixations between the Target and Competitor objects and the Unrelated object was smaller. But as we did not find an interaction between condition and task order, we dropped the interaction term and considered whether fixations depended on condition in the consecutive interpreting task. Participants began looking significantly more at the target object from -550ms before word onset until at least 1000ms after onset of the predictable word. There was no significant difference in fixations on the English competitor and the unrelated object over a period of at least three bins, nor any trend in this direction. Thus, participants made predictive fixations on target objects and persisted in these fixations in both the consecutive and listening tasks, but we did not find evidence of phonological prediction, and we found evidence of phonological activation only in the listening task.

We directly compared the listening and consecutive tasks using a linear mixed model and specifying an interaction of task for each condition (listening/consecutive). There was no significant interaction of task with target vs. unrelated condition at any point. Although there was a significant interaction of task with competitor vs. unrelated condition in the period at 850ms, this interaction did not take place over a sustained period of at least three consecutive bins [see 4; 73]. We thus conclude that there is no evidence of a task-dependent significant difference in fixation proportions.

**Noisy Speech study—Filler items.** The differences between the target and unrelated conditions could be due to differences in the visual properties of the pictures (e.g., colour, animacy) and hence, how much they attracted attention. If so, we would expect to see differences in fixation proportions on target and unrelated conditions when they were presented with neutral, filler sentences as well. We first checked this for the listening task with a model that also included task order. However, there was no interaction at any point between task order and condition. Therefore, we dropped the interaction term from the model and considered whether predictive fixations depended on condition. Predictive fixations depended on condition from -450ms to 0ms, suggesting a visual bias towards the competitor object for the listening task (Fig 4). For the consecutive task, there was an interaction between condition (target) and task order at 400ms and an interaction between condition (competitor) and task order from 100 to 200ms. However, the interaction was not significant consistently or over at least three bins in a row. We therefore dropped the interaction term from the model, and ran a model that considered whether predictive fixations depended on condition. Here, we did not find any significant differences between the target and unrelated or competitor and unrelated conditions at any time point (Fig 4).

The series of significant differences between the competitor and the unrelated condition in the listening task could suggest that any predictive fixations on competitor objects in the experimental sentences were due to visual bias (rather than prediction). However, we did not find any predictive fixations on competitor objects in the experimental items. On the other

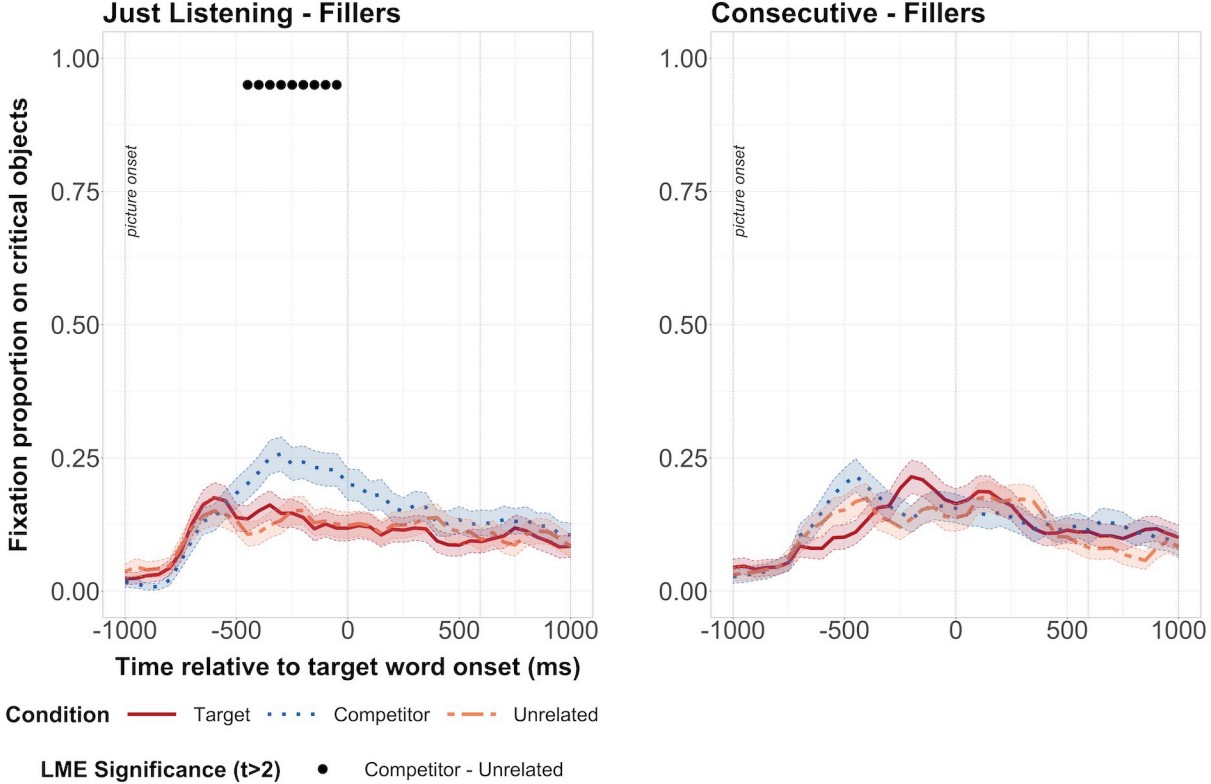

**Fig 4. Fixation proportions for the filler items in the in the Noisy Speech study.** The listening task is shown on the left and the consecutive task on the right. Fixation proportions on target, competitor and unrelated objects when images were presented with the filler sentences. Black dots along the top represent bins during which there was a significant difference between competitor and unrelated conditions. Transparent thick lines are error bars representing standard errors.

hand, we did not find any evidence of visual bias towards the target image in the filler items, so we can be confident that the differences between target and unrelated conditions in the experimental items were not due to uncontrolled differences in the images.

**Clear Speech study—Experimental items.**   We then investigated the eye-tracking data from the Clear Speech study. In the listening task, participants began fixating the target object at -400ms until at least 1000ms after the predictable word onset (Fig 5). Participants also fixated on the competitor, albeit non-predictively from 700 to 950ms. In the consecutive task, participants began fixated the target object from -500ms before predictable word onset until at least 1000ms after predictable word onset (Fig 5). They also fixated the competitor at several different points, at -200ms, from 300-400ms and from 800-900ms. However, there was no significant difference between the competitor and the unrelated condition that lasted for at least three consecutive bins. Thus, participants predicted the upcoming word in both the listening and consecutive tasks, but did not form any phonological predictions. In the listening task, participants fixated the phonological competitor after predictable word onset, suggesting phonological activation.

We then compared the listening and consecutive tasks using the same linear mixed model as for the Noisy Speech study. We did not find any significant interactions between task and either the target or the competitor (vs. unrelated) conditions. Thus there is no evidence that the consecutive interpreting task affected fixation proportions on the target or competitor object.

**Clear Speech study—Filler items.**   We then considered the filler items for the just listening task in the Clear Speech study. As in the other analyses, we first checked whether the order in which participants carried out the two tasks interacted with the proportion of fixations on

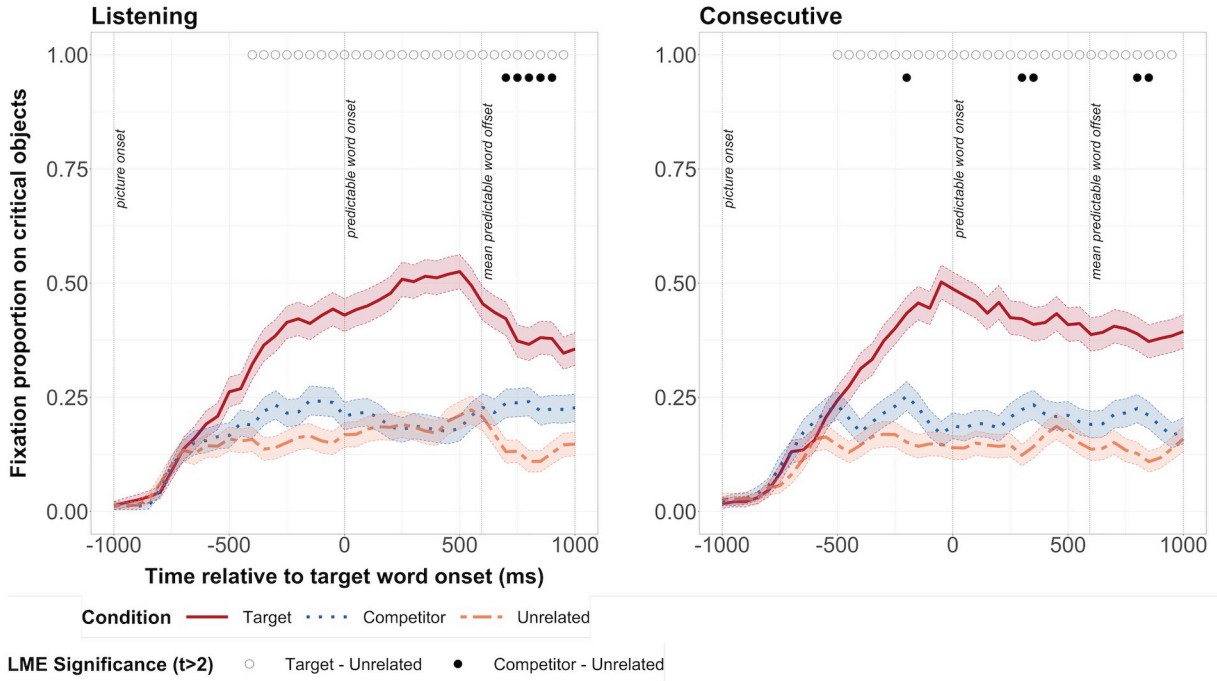

**Fig 5. Graph showing fixation proportions on target, competitor and unrelated objects in the Clear Speech study.** The listening task is shown on the left, and the consecutive task on the right. Open circles along the top represent bins during which there was a significant difference between fixation proportions on target and unrelated objects. Filled circles represent bins during which there was a significant difference between fixations proportions on competitor and unrelated objects. Transparent thick lines are error bars representing standard errors.

target and competitor vs. unrelated objects. We found that between 150 and 300ms after word onset, target vs. unrelated fixations depended on task order, with a greater difference between fixations on the target vs unrelated object when participants did the listening task first. Two time bins were also significant for the interaction between the difference in competitor vs. unrelated fixations and task order, again with a greater difference in fixations on competitor vs. unrelated object when participants carried out the listening task first. In both cases, there were more fixations on the unrelated object compared to the target/competitor object. As we did not find any interaction during the predictive time window (before word onset at 0ms) we ran the analysis again without the interaction term. We did not find any significant differences between fixations on the target or competitor vs. unrelated object (Fig 6).

We ran the same analysis for the filler items in the consecutive task. We did not find any significant interactions between task order and fixation proportions on critical objects. Again, we dropped the interaction term and considered whether predictive fixations depended on condition. We found a significant difference between fixations on the unrelated and target objects, with fewer fixations on the target as compared to the unrelated object over three bins from -150ms to 0ms. This goes in the opposite direction from our results from the experimental items.

Based on these analyses of the filler items, we are able to exclude visual bias as a reason for the predictive fixations on the target objects that we found in the experimental items.

**Comparison of Noisy and Clear Speech studies.** Our main analyses showed that L2 listeners predict in noisy listening conditions, just as they predict in clear listening conditions. In both the Noisy and Clear Speech studies, a significant divergence between fixations on target

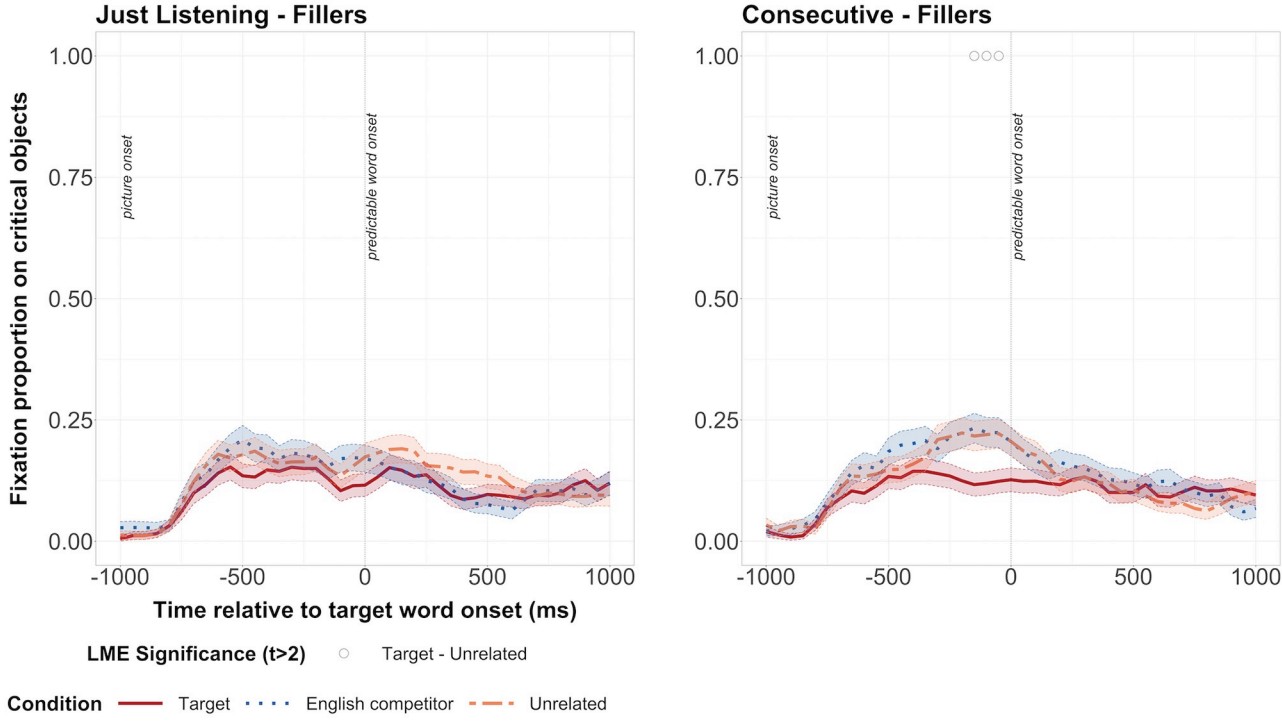

**Fig 6. Fixation proportions for the filler items in the in the Clear Speech study.** The listening task is shown on the left and the consecutive task on the right. Open circles along the top represent bins during which there was a significant difference between target and unrelated conditions. Transparent thick lines are error bars representing standard errors.

and unrelated objects began from between -550ms to -400ms before onset of a predictable word, and we did not find any evidence to suggest that such fixations depended on task.

However, by analysing the two studies separately, we cannot establish whether there was a difference in the magnitude of the prediction effect between the Noisy and the Clear Speech studies. We thus ran a model to establish whether noisy speech affected predictive processing in the listening and consecutive tasks. Our model took arcsine transformed fixation proportions as a dependent variable and contained a three-way interaction term for condition (target/competitor/unrelated), speech type (noisy/clear) and task (consecutive/listening). We did not find any sustained evidence of a three-way interaction. We found that in two sets of two bins after word onset (from 300–400 and from 450 to 550ms) the difference between the competitor and unrelated condition depended on both speech type and task. However, this was not a reliable difference over a series of at least three time bins. We did however find that there was a significant difference in fixation proportions that depended on an interaction between noise and competitor over three time bins from 750 to 900ms. We therefore dropped task as a predictor variable in our model, and ran a model that compared fixation proportions in noisy vs. in clear speech (with condition and speech type as our independent variables). In this model, we did not find any effect of noise, either as a main effect or as an interaction effect. We did, however, find a main effect of phonological competitor effect over three time bins, from -200ms to -50ms when we considered the data in this way. Therefore, we dropped the interaction term from the model and ran our model with only condition as a predictor over all data. We found a significant divergence in fixations between the target and unrelated conditions from -550ms until 1000ms after target word onset (Fig 7). We also found a significant

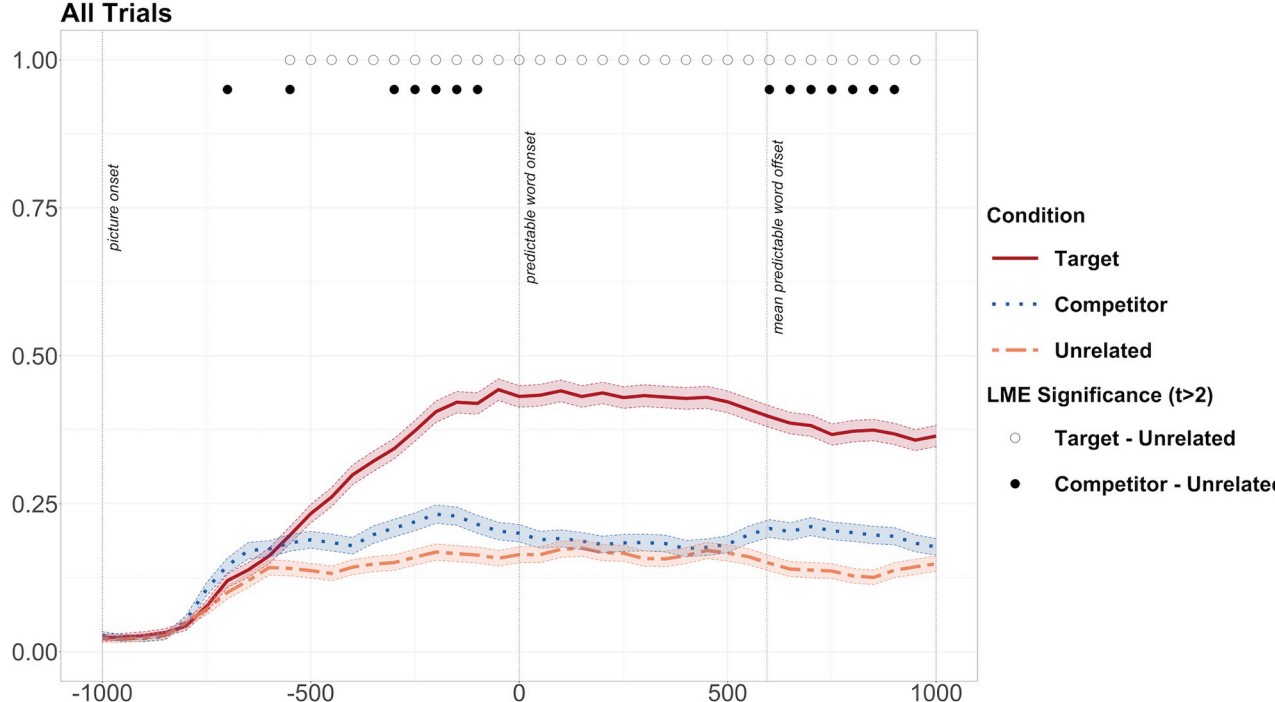

**Fig 7. Graph showing fixation proportions for all experimental trials across both Noisy and Clear speech studies.** Open circles along the top represent bins during which there was a significant difference between target and unrelated conditions. Black dots along the top represent bins during which there was a significant difference between competitor and unrelated conditions. Transparent thick lines are error bars representing standard errors.

divergence between the competitor and unrelated conditions between -300ms and -100ms and from 600ms to 950ms.

However, as we had previously found evidence of visual bias towards the English competitor object in the predictive window (before 0ms) when the objects were presented with a neutral sentence in the Noisy Speech study in the listening condition, we checked whether this visual bias was also present across the study as a whole, and whether this could account for the competitor effect. We found that fixations on the competitor and unrelated objects diverged from -450ms to -250ms and in one isolated bin at 600ms (see Fig 8). Target vs. unrelated objects diverged once at 400ms, with fewer fixations on the target object. The pattern of predictive fixations on competitor items in filler/experimental trials was not exactly the same, as predictive fixations in the experimental trials occur later in the predictive window. However, we cannot exclude the possibility that predictive fixations in the experimental trials are the result of visual bias to the competitor object. However, we did not see a competitor effect after the onset of the predictable word, as we did in the experimental trials. Therefore, we can be confident that the sustained divergence in fixations on competitor vs. unrelated objects from just after the mean onset time of the predictable word at 600ms is not due to visual bias.

**Correlation analyses.** Finally, we considered whether our null results were linked to certain characteristics of our participants. Firstly, some of the participants in our study had received minimal training in consecutive interpreting. It may be that people only employ more top-down listening strategies in a consecutive interpreting task after training. We thus assessed whether training in consecutive interpreting was correlated with the extent of predictive fixations on target and competitor objects as compared to unrelated objects. Here we considered all of the data from both the noisy and the clear speech studies together. We computed the

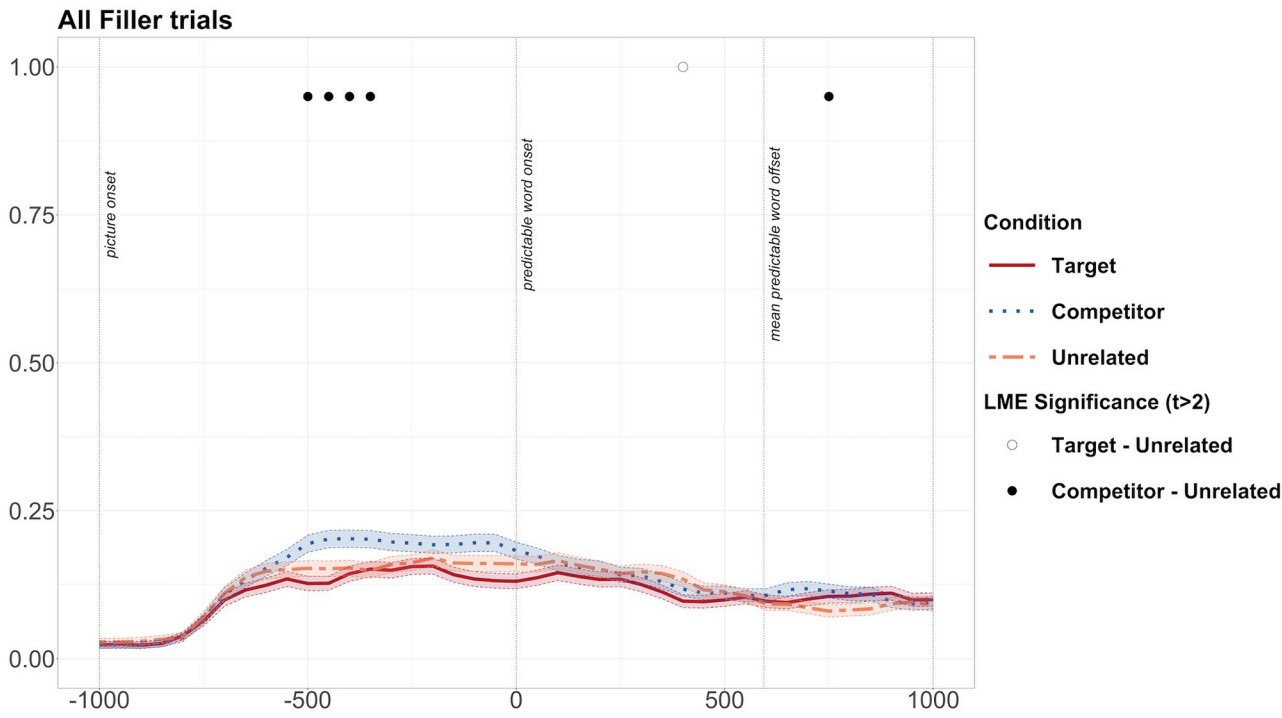

**Fig 8. Graph showing fixation proportions for all filler trials across both Noisy and Clear speech studies.** Fixation proportions on target, competitor and unrelated objects when images were presented with the filler sentences. Black dots along the top represent bins during which there was a significant difference between competitor and unrelated conditions. Transparent thick lines are error bars representing standard errors.

difference, during the consecutive interpreting task, in the mean arcsine-transformed fixation proportions between the target and unrelated conditions in the window during which we found evidence of prediction, from -550ms before word onset and until word onset. -550ms was the earliest time at which predictive fixations began in the consecutive interpreting task. Fig 9 shows that the extent of predictive fixations on the target object did not correlate with the number of months of training that participants had received ($r(70) = .09$, $p = .46$). The same was true of predictive fixations on the competitor object ($r(70) = —.08$, $p = .52$).

Secondly, participants' L2 proficiency could have affected the results. On the one hand, phonological prediction might be more likely if participants' L2 proficiency is higher. On the other hand, semantic prediction in noise might be less likely if participants' L2 proficiency is lower. We therefore considered whether predictive fixations in both tasks were related to L2 proficiency as measured by the participants' LexTale scores. Here again we computed the difference in the mean arcsine-transformed fixation proportions between the target and unrelated conditions and competitor and unrelated conditions in the window from -550ms (the start of predictive eye movements) before word onset and until word onset. We did not find a significant correlation between fixations on the competitor object in the predictive window linked to participants' level of English proficiency ($r(70) = .07$, $p = .53$). However, we did find a moderate, significant correlation between participants' level of English proficiency as measured by LexTale score and the extent of their predictive fixations on the target, as opposed to

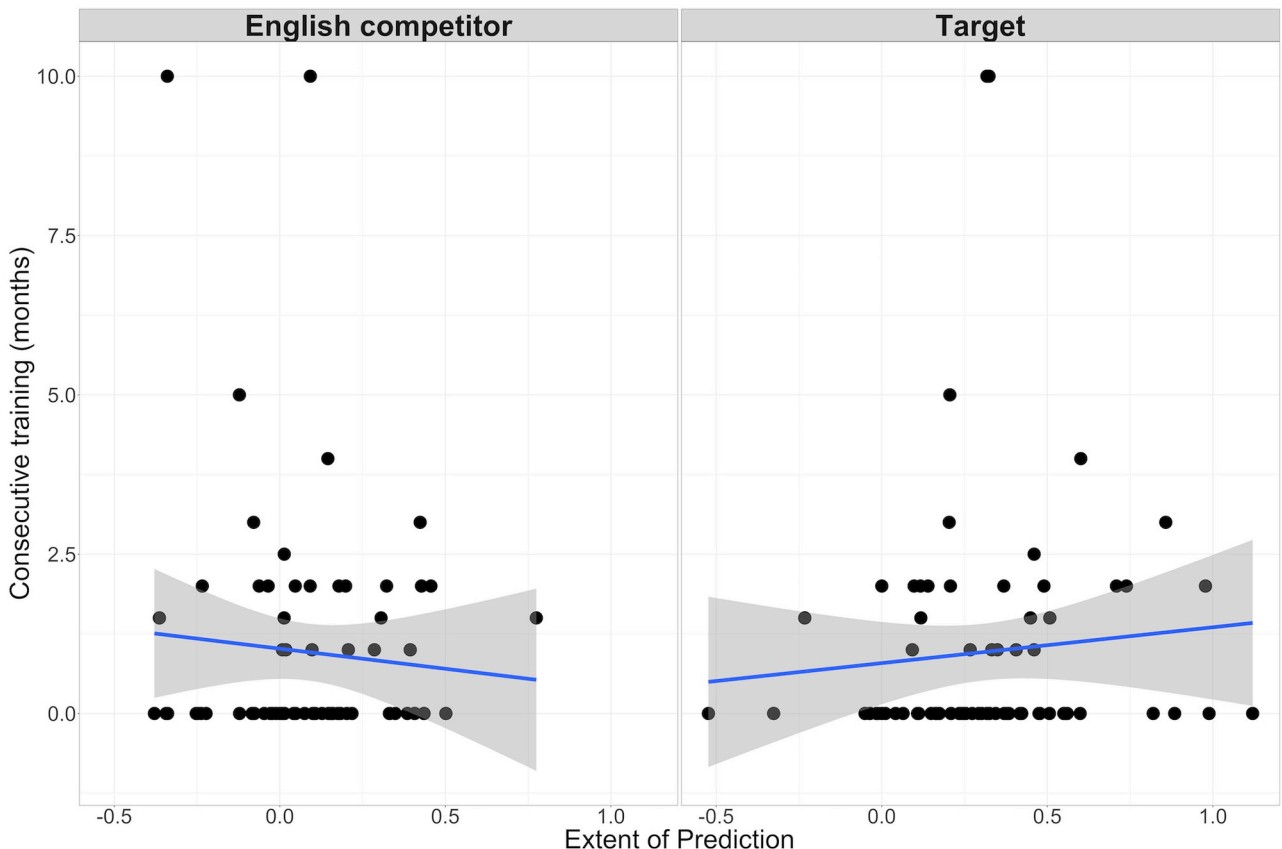

**Fig 9. The (lack of) relationship between training in consecutive interpreting and the extent of prediction.** The y-axis shows the number of months of consecutive interpreting, and the x-axis shows the difference in fixation proportions between the Unrelated and the English competitor (left) and Target (right) conditions.

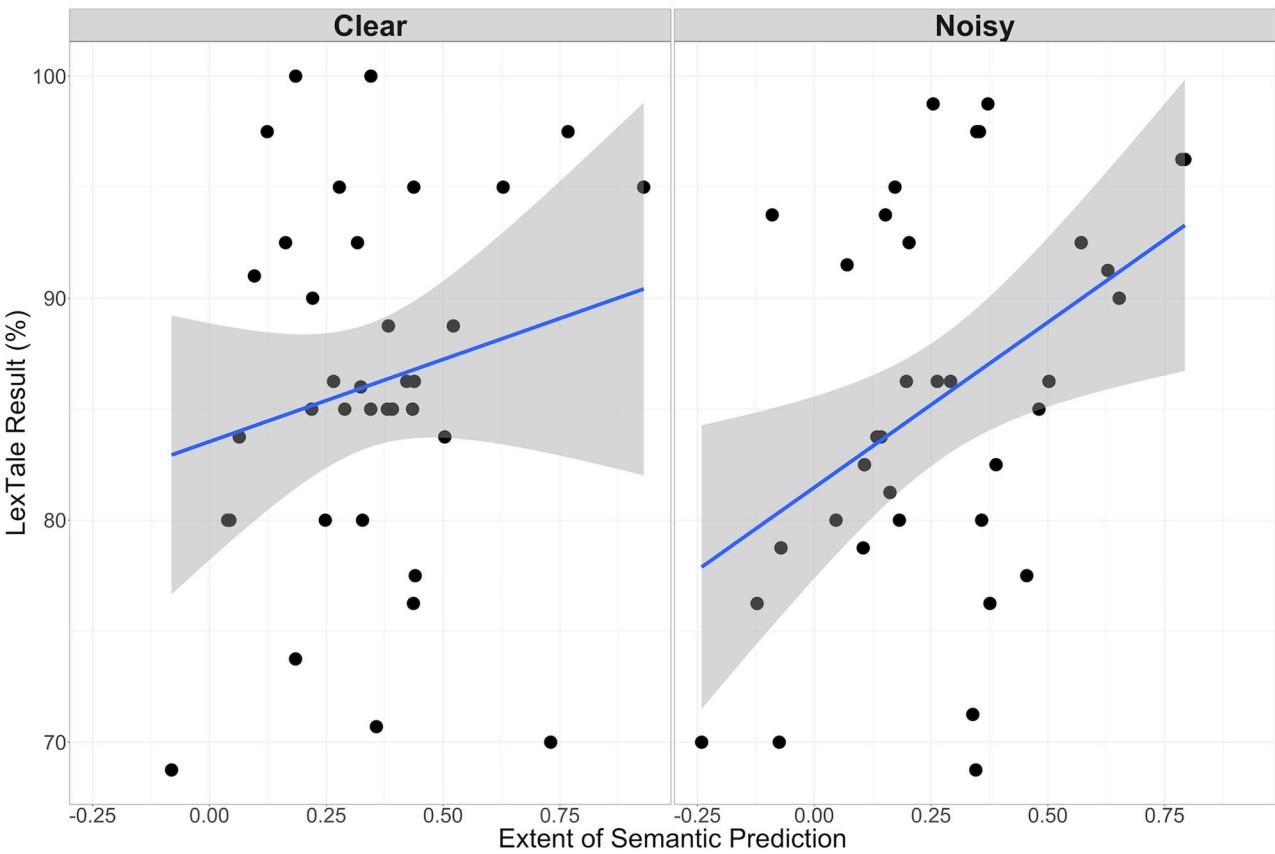

**Fig 10. The relationship between participants' Lextale result and the extent of semantic prediction.** The y-axis is the LexTale result, expressed as a percent, and the x-axis is the difference in fixation proportions between target and unrelated objects in the period from -550ms before word onset until word onset. Results from the Clear Speech study on the left, and from the Noisy Speech study on the right.

the unrelated object ($r(70) = .32$, $p = .007$). We checked whether this correlation existed in both the Noisy and the Clear Speech studies. We found that the correlation was present in the Noisy Speech study ($r(34) = .42$, $p = .01$), but not in the Clear Speech study ($r(34) = .19$, $p = .27$) (Fig 10).

Raw data and scripts for these analyses are available on Open Science Framework at: https://osf.io/5dfmr/.

## Discussion

### L2 listeners make semantic predictions in speech-shaped sound

Our findings demonstrate that L2 listeners predict upcoming utterances both in speech-shaped sound and in clear speech conditions. This extends previous findings showing that L2 listeners make predictions in clear speech conditions [4, 10, 13, 76]. In order to make semantic predictions, L2 listeners use contextual and lexical information. It therefore appears that at a moderate level of energetic masking (SNR of 0 dB), when speech is still intelligible, L2 listeners continue to rely on top-down processing during comprehension. This result contradicts findings suggesting that L2 listeners do not make use of contextual information while listening in noise [e.g., 35], because L2 listeners made predictive eye movements when listening to highly predictable sentences. Our study thus demonstrates that semantic prediction in L2 is robust

enough to take place even in challenging conditions (specifically, energetic masking) and that L2 listeners engage in top-down processing in energetic masking. This in turn suggests either that if cognitive resources are being used to process noisy speech, L2 listeners still have sufficient resources for prediction, or alternatively that semantic prediction does not require cognitive resources.

## No evidence that a consecutive task affects prediction

We did not find any significant difference in predictive fixations between the listening and consecutive tasks in either the noisy speech or the clear speech study. Our analysis revealed very similar fixation patterns for both tasks in both studies in the predictive time window: in the Noisy Speech study, fixations on the target and unrelated objects diverged at -550ms for the consecutive and at -450ms for the listening task; in the Clear Speech study, fixations on the target and unrelated objects diverged at -500ms for the consecutive and at -400ms for the listening task. We did not find consistent evidence of a divergence between the competitor and unrelated conditions in either task in either study. Thus, asking participants to listen to utterances in order to consecutively interpret them, or simply asking them to listen to utterances, did not lead to any difference in predictive processing. Contrary to Zhao et al. [58], we did not find that engaging the production mechanism during a consecutive interpreting task influenced prediction. Therefore, we found no evidence that predictions are made strategically during consecutive interpreting, or that focusing attention influences prediction. However, there are other possible explanations for these null results which we detail below.

One reason for this lack of difference could be that our stimuli were individual sentences with simple (and predictable) content. The design of our experiment led to a consecutive interpreting scenario most similar to "short consecutive" [42], when interpreters interpret one sentence at a time. The fact that participants interpreted individual sentences, which were relatively short and simple, and were not lexically dense, might have meant that the effort required to interpret them consecutively was very similar to the effort required to listen to them. Had the sentences been longer, more complex and denser, there might have been a difference between the consecutive and the listening tasks.

Equally, it may be that only interpreters who are fully trained in consecutive interpreting use more top-down listening strategies when carrying out a consecutive interpreting task as compared to just listening. Setton and Dawrant [45] suggested that the skills required for active listening should be acquired during training in consecutive interpreting, and thereby assumed that the consecutive interpreting task does not, in itself, lead students to engage in active, focused listening. Our correlational analysis does not show any link between (minimal) training and prediction during a consecutive interpreting task. However, we cannot exclude the possibility that additional training and experience would have an effect.

Finally, it is possible that participants did not consistently plan their utterance during the consecutive interpreting task. In conversation, gaps between interlocutors' speech are usually around 200ms (though with a lot of variability) [77], while in this study, the mean gap between offset of the spoken sentence and onset of the interpreted sentences was 1148ms or 1146ms for Noisy and Clear Speech respectively—around a second longer; note, however, that this was subject to a lot of variability (Noisy Speech SD: 954ms, Clear Speech SD: 789ms), perhaps due to differences in the lengths of sentences. Based on this, it seems possible that participants first just listened to the speech stream, and only then planned their own utterance. When speech planning starts substantially after sentence offset, a consecutive interpreting task may not lead to greater engagement of the production mechanism than a task that just involves listening.

## No evidence of word-form prediction

Although participants made predictive eye movements towards an object depicting the predictable word, we did not find reliable evidence that they predictively activated the phonological form of the predictable word. This is consistent with findings from previous studies with L2 participants [4, 9, 12]. When we analysed the two studies separately, we found no evidence that L2 listeners engage in phonological prediction in noisy conditions, even though such predictions might be particularly beneficial, and even though the production mechanism could support such predictions during the consecutive task. According to a prediction-by-production account, where phonological prediction would follow semantic prediction, it may be that predictions in L2 in noise are slowed down such that semantic prediction takes place but phonological prediction does not. It is also possible that cognitive resources are required to make phonological predictions, and that sufficient resources were not available to participants listening to their L2 in noise. When we analysed both studies together, we did find a phonological competitor effect before onset of the predictable word that did not depend on task. However, we also found an effect of visual bias at a similar time period in the filler sentences, so we cannot consider this evidence as reliable.

We did however find evidence of phonological activation after participants heard the critical word in the listening tasks in both the noisy and the clear speech studies, and when we considered the findings of both studies together across tasks. This effect took place at around the mean offset time of the predictable word. Thus, our studies show that phonological activation did take place. While some studies have found more sustained fixations on phonological competitors in noisy than in quiet conditions, suggesting that noise makes it difficult to disambiguate between target words and phonological competitors [68], we did not find convincing evidence of this in our study because there was no significant interaction between the competitor condition (vs. unrelated) and the speech type (noisy or clear).

We did not observe any late competitor effect in the consecutive interpreting task in either study. This could be because listeners were additionally preparing to consecutively interpret by activating translation equivalents of English words in Dutch to prepare their utterance, and were therefore less sensitive to the phonological overlap in English. However, we also did not find any significant interaction between task and condition (target or competitor vs. unrelated).

## No evidence that Noisy Speech affects prediction in L2 users

We did not find evidence of different predictive patterns in L2 listeners in the Noisy vs. the Clear Speech study. This suggests that not only does semantic prediction take place in noise, but also that noise may not affect the extent of prediction. However, we did find a correlation between the level of proficiency of L2 speakers and the extent of prediction during comprehension. When we broke down this result by study, we found that there was a correlation between level of proficiency in the Noisy Speech study, but not in the Clear Speech study. This suggests that, in noisy speech, the level of L2 proficiency may be linked to the extent of prediction during comprehension.

## Conclusions

We reported an experiment that investigated whether L2 listeners make predictive eye movements when comprehending in noisy listening conditions, and whether subsequently producing an utterance (i.e., in a consecutive interpreting task) influences prediction. We found that L2 listeners do make predictive eye movements in noise, but that a consecutive interpreting task does not have a clear effect on patterns of predictive processing in noise. Thus, L2 listeners

predict, even with a noisy background. In addition, we did not find evidence that listening in noise differed significantly from listening in quiet. This finding supports theories of prediction in which semantic prediction takes place before phonological prediction or uses a different mechanism from phonological prediction. It also demonstrates that semantic prediction takes place even in communicative settings that may be found outside the experimental laboratory and leaves open the possibility that phonological prediction may require cognitive resources. Our study also opens up various avenues for future research. For instance, future studies of prediction during comprehension could investigate whether and how noise affects prediction during consecutive interpreting at a range of signal-to-noise ratios. Other studies might consider whether and how noise affects semantic and phonological prediction in L1 and L2. Additionally, the level of proficiency in L2 could be considered, to determine the minimum level of proficiency required by L2 users to make predictions during comprehension. Meanwhile, studies of language production could consider whether prediction during consecutive interpreting affects the speed and quality of subsequent production.

## Supporting information

**S1 Appendix. Experimental items.**
(DOCX)

## Acknowledgments

We would like to thank Dr. Mieke Slim, who recruited the participants for our norming studies and Dr. Matthias Franken, who advised us on how to create speech-shaped sound.

## Author Contributions

**Conceptualization:** Rhona M. Amos, Robert J. Hartsuiker, Kilian G. Seeber, Martin J. Pickering.

**Data curation:** Rhona M. Amos, Robert J. Hartsuiker.

**Formal analysis:** Rhona M. Amos.

**Funding acquisition:** Rhona M. Amos.

**Investigation:** Rhona M. Amos.

**Methodology:** Rhona M. Amos, Martin J. Pickering.

**Project administration:** Robert J. Hartsuiker, Kilian G. Seeber, Martin J. Pickering.

**Supervision:** Robert J. Hartsuiker, Kilian G. Seeber, Martin J. Pickering.

**Validation:** Robert J. Hartsuiker, Martin J. Pickering.

**Visualization:** Rhona M. Amos.

**Writing – original draft:** Rhona M. Amos.

**Writing – review & editing:** Robert J. Hartsuiker, Kilian G. Seeber, Martin J. Pickering.

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
