## [Decision Letter · Decision Letter 0]

22 Aug 2022

PONE-D-22-18894Purposeful listening in challenging conditions: a study of prediction during consecutive interpreting in noisePLOS ONE

Dear Dr. Amos,

Thank you for submitting your manuscript to PLOS ONE. After careful consideration, we feel that it has merit but does not fully meet PLOS ONE’s publication criteria as it currently stands. Therefore, we invite you to submit a revised version of the manuscript that addresses the points raised during the review process.

I have received the comments from two experts in the fields. As you will see, both of them rise several issues both at methodological (the majority) and theoretical level. The two reviews are very detailed and rich of very good suggestions, which I also share with them. If you decide to proceed with the revision, I encourage you to carefully consider all the reviewers' comment, which will require extensive work, also in terms of data collection.  

We look forward to receiving your revised manuscript.

Kind regards,

Simone Sulpizio

Academic Editor

PLOS ONE

Journal Requirements:

a) Did participants provide their written or verbal informed consent to participate in this study?

Reviewers' comments:

Reviewer's Responses to Questions

**Comments to the Author**

1. Is the manuscript technically sound, and do the data support the conclusions?

Reviewer #1: Partly

Reviewer #2: Yes

2. Has the statistical analysis been performed appropriately and rigorously? 

Reviewer #1: No

Reviewer #2: Yes

3. Have the authors made all data underlying the findings in their manuscript fully available?

Reviewer #1: Yes

Reviewer #2: Yes

4. Is the manuscript presented in an intelligible fashion and written in standard English?

Reviewer #1: Yes

Reviewer #2: Yes

5. Review Comments to the Author

Reviewer #1: The paper reports an eye-tracking experiment to test predictive processing in noisy circumstances in proficient L2 speakers, also testing for word-form prediction and sustained attention by manipulating the participants’ task while performing the eye-tracking visual world experiment.

One of the questions that is addressed is whether L2 speakers rely on prediction during sentence processing in noisy (more challenging) conditions.

The idea is that, while L1 speakers might make use of top-down processing to repair the noise, L2 speakers might not be able to do so, due to limited cognitive resources (processing in the L2 is more demanding, per se) that would prevent L2 speakers to make use of contextual cues to predict the upcoming words.

The result will tell us about the processes involved in prediction: which aspects are blocked in noisy circumstances, and which are maintained? Specifically, the authors are testing predictions which rely on the phonological similarity of the uttered words (e.g., mouth/mouse).

Another aspect that the authors are considering is how the task influences the processing. To this purpose, they administered two different tasks to participants (in two blocks - within subject design): a simple listening task and a consecutive interpreting task in which people are asked to repeat the sentence in their L1 after listening to the sentences in the L2.

The idea is that in the consecutive interpreting task more focused attention is required, and this might enhance prediction. Moreover, this type of task might also favor word-form prediction (because planned production is involved), thus possibly increasing looks to the phonological competitor compared to the (plain) looking-while-listening task (also, an effect of the speed of the produced translation).

The results of the main analyses are the following:

1. L2 listeners are able to predict in noisy conditions, thus they do on top-down processes during sentenceprocessing/comprehension.

2. No reliable evidence of phonological prediction or activation in any of the two tasks

3. no task-dependent significant difference in fixation proportion towards the target or competitor vs. unrelated picture.

Additional analyses are reported:

4. An exploratory analysis on translation onset revealed that when interpretation starts later, prediction is more robust (more looks to the target and the competitor vs. unrelated object).

The authors also carry out additional analyses to check for (5) task order (significant effects), (6) visual bias of the pictures (no effect), (7) participants’ training/experience (no significant correlation), (8) participants’ level of L2 (no significant correlation).

As for conclusion in (1), the authors explain this result by concluding that processing in the L2 in challenging circumstances leaves enough resources to make use of top-down processes.

However, as pointed out by the authors themselves, this might be explained by the fact that semantic prediction is not cognitively demanding per se.

I do not think that this study can offer a response to this question: first of all, the population investigated in this study is high proficient in the L2 (the non-significant correlations between training experience or lextale scores might simply depend on the small sample, and to the little variability within the sample itself) ; second, cognitive resources were not tested/modulated in this study, thus it is unclear if, or how, they contributed to the effect.

So, while there is evidence for prediction, it is not clear if this is due to high performance in the L2 or to the cognitive resources involved in the process, or to other factors. And I do not think that anything can be concluded with respect to this issue.

Furthermore, I am not convinced that the type of task implemented in this study requires additional cognitive resources (as assumed by the authors): if it is true that it is challenging, due to the introduced noise, it might simply disrupt the standard process for lexical access (via phonological cues), thus demanding for more top-down processes (based on semantic cues) to repair the noise (which in turn might be linked to language proficiency, not cognitive resources).

In some passages, the authors seem to assume that prediction is costly, in contrast with the standard assumption that prediction speeds up processing (and it has been also shown for children). Please clarify.

In this respect, I would be more cautious in the conclusions, for example when the authors state “It also demonstrates that semantic prediction takes place even when cognitive resources are limited, and leaves open the possibility that phonological prediction may require cognitive resources.”

As for the null results reported: although in general I do favor reporting null results, I am skeptical that these are reliable in this case, due to the little sample tested and the many manipulations introduced/conditions tested. Although the authors argue for the reliability of their participants’ sample size (in line with previous reported studies) I believe that testing less than 30 participants with eye-tracking measures do not provide enough statistical power to gain reliable results, especially in the case post hoc analyses are conducted on a limited number of datapoints, as it is the case here.

Furthermore, I wonder whether the null result obtained in comparing the two tasks might be due to the background of the participants: being trained as interpreters, they might approach the listening task in an “interpreter” mode, and this might have obscured a potential difference.

In general, I wonder why the factors tested in the additional analyses were not introduced as predictors or random factors in the main models. I think this would make the data treatment clearer and more robust.

Also, I think that a control group of L1 English speakers and/or more variability within the L2 speakers (with respect to their proficiency in the L2) is needed to get a baseline and answer some of the questions addressed in the paper.

Overall, I find the paper quite difficult to follow, due to the many questions addressed at the beginning (most of which remain, in fact, without a robust answer), and the different main and additional analyses presented, most of which did not return significant results.

I suggest either focusing only on the main question addressed (do (proficient) L2 speakers rely on prediction during sentence processing in noisy (more challenging) conditions?) or expanding the groups so as to increase statistical power, also including L1 speakers and/or L2 speakers with more variable linguistic (and training) background.

Other points:

- I suggest changing the example in the paper, since this particular example contains a flow in the stimuli: it is not only a “mouth”, it is an “open mouth”, thus the cue is double in this case (dentist primes mouth; open primes an open (mouth)). As far as I can tell from the list of stimuli in the appendix this doesn’t apply to every example (although in other cases as well there is a clear double (or even triple) cue pointing to the target: e.g., 10. honey + stung  bee; 18. Student +library +read book

- What about cross-linguistic competitors (in Dutch?). Nothing is said about this, was this controlled for in selection of the stimuli? (for example, both the Dutch translations of mouth and mouse begin with an “m”; bed and ladder in English are very similar to their Dutch translations, but the phonological English competitors are not)—given that Dutch is specifically involved in one of the tasks, these similarities might introduce an additional (not controlled?) confound.

- Given the very small set of stimuli presented, I wonder whether seeing the same array of objects in the visual display (albeit in different positions) had an impact on the participants’ performance. We know form other studies that even young children DO remember what has been shown to them before! This is not standard practice in visual world experiment - to control for visual salience typically a Latin square design is employed and the same array of objects is shown in different conditions between, not within subjects. Please clarify this choice.

Reviewer #2: This paper presents evidence that people predict upcoming words when listening to their second language in noisy conditions. In an eye-tracking experiment using the visual world paradigm, participants looked to target images, corresponding to upcoming English words, prior to the onset of those words in high-cloze sentences. The participants were high-proficiency L2 English speakers, and the experiment manipulated whether they listened for comprehension or interpreted those sentences into their L1 (Dutch), manipulated within subjects in two blocks. This consecutive interpreting task did not affect fixation on target images, nor were participants more likely to look to phonological competitors compared to unrelated competitors. However, participants who fixated on target images earlier were slower to begin interpreting, and participants who performing the interpreting task in the first block (i.e., listened only for comprehension in the second half of the experiment) fixated on phonological competitor images more than participants who interpreted in the second block did.

This paper replicates previous findings that people predict upcoming words even in their L2, and it builds on that finding by demonstrating that such prediction occurs even in noise. Given the null effect of interpretation on fixation (and the null effect of fixation on phonological competitors vs unrelated competitors), the key question seems to be whether adding noise during listening is a substantial novel contribution. Having listened to some of the audio files kindly made available on OSF, I’m skeptical; the experience is much like listening to someone speaking quietly. Without a manipulation of the noise condition (e.g., energetic vs informational masking, or varying the SNR), it is hard to contrast these results with L2 listening in previous studies, especially given that the participants in this experiment were high-proficiency speakers of a closely related language (unlike Ito et al., 2018, and Zhao et al., 2022).

The sample size (N = 24) is also concerning, especially because the experiment was not pregregistered. The authors based their sample size on Ito et al. (2018), but that study recruited a total of 48 participants (albeit two groups of 24, only one of which comprised L2 speakers, with 16 items). This experiment added another (within-subject) manipulation and included 30 items, for (I believe) a total of 120 observations per condition. It is almost certainly underpowered. I appreciate the difficulty of recruiting from a specialized population (L1 Dutch interpreters), but the exploratory analyses require replication before they can be accepted with any confidence. The headline effect of fixation on target images is more robust, but again, neither L2 status nor amount/type of noise was manipulated in the design.

I am also curious about the presentation of the visual stimulus at 1s prior to the onset of the target word. To my knowledge, this is different from common VWP practice where the visual stimulus is presented prior to the onset of the sentence. I would like to author to justify the divergence of their setup from the common practice. Given the sentences, it is likely that the visual stimulus was presented AFTER the sentence started, which may discourage people from looking too much at the pictures (which may then drown any phonological effect they were looking for). Also, I think it is also important to describe when the pictures were presented relative to the onset of the sentence, given that the target word might occur in different positions in the sentence.

I recommend that the authors conduct a follow-up experiment to increase confidence in the exploratory findings and/or to better demonstrate the impact of noise on L2 prediction. As it stands, the novelty of adding energetic masking during L2 listening is questionable, and the sample size is too small to conclude that consecutive interpreting does not affect prediction or that the exploratory effects will replicate.

6. PLOS authors have the option to publish the peer review history of their article (what does this mean?). If published, this will include your full peer review and any attached files.

Reviewer #1: No

Reviewer #2: No

---

## [Author Response · Author response to Decision Letter 0]

15 May 2023

We would like to thank the editor and the reviewers for the careful consideration of our first manuscript. We have addressed the methodological and theoretical issues raised by the reviewers and highlighted by the editor. In particular, we have carried out extensive data collection work. We increased the number of participants in our original study by 50% (from 24 to 36) and carried out a further study in clear speech conditions, again with 36 participants. We have also extensively revised the manuscript, by streamlining the statistical analyses and focusing on the main research questions. Below, please find our responses to individual comments. We refer to the line numbers we provided in the clean version of our revision (file name: Manuscript). For clarity, our responses are in blue.

Reviewer #1: The paper reports an eye-tracking experiment to test predictive processing in noisy circumstances in proficient L2 speakers, also testing for word-form prediction and sustained attention by manipulating the participants’ task while performing the eye-tracking visual world experiment.

One of the questions that is addressed is whether L2 speakers rely on prediction during sentence processing in noisy (more challenging) conditions.

The idea is that, while L1 speakers might make use of top-down processing to repair the noise, L2 speakers might not be able to do so, due to limited cognitive resources (processing in the L2 is more demanding, per se) that would prevent L2 speakers to make use of contextual cues to predict the upcoming words.

The result will tell us about the processes involved in prediction: which aspects are blocked in noisy circumstances, and which are maintained? Specifically, the authors are testing predictions which rely on the phonological similarity of the uttered words (e.g., mouth/mouse).

Another aspect that the authors are considering is how the task influences the processing. To this purpose, they administered two different tasks to participants (in two blocks - within subject design): a simple listening task and a consecutive interpreting task in which people are asked to repeat the sentence in their L1 after listening to the sentences in the L2.

The idea is that in the consecutive interpreting task more focused attention is required, and this might enhance prediction. Moreover, this type of task might also favor word-form prediction (because planned production is involved), thus possibly increasing looks to the phonological competitor compared to the (plain) looking-while-listening task (also, an effect of the speed of the produced translation).

The results of the main analyses are the following:

1. L2 listeners are able to predict in noisy conditions, thus they do on top-down processes during sentenceprocessing/comprehension.

2. No reliable evidence of phonological prediction or activation in any of the two tasks

3. no task-dependent significant difference in fixation proportion towards the target or competitor vs. unrelated picture.

Additional analyses are reported:

4. An exploratory analysis on translation onset revealed that when interpretation starts later, prediction is more robust (more looks to the target and the competitor vs. unrelated object).

The authors also carry out additional analyses to check for (5) task order (significant effects), (6) visual bias of the pictures (no effect), (7) participants’ training/experience (no significant correlation), (8) participants’ level of L2 (no significant correlation).

As for conclusion in (1), the authors explain this result by concluding that processing in the L2 in challenging circumstances leaves enough resources to make use of top-down processes.

However, as pointed out by the authors themselves, this might be explained by the fact that semantic prediction is not cognitively demanding per se.

I do not think that this study can offer a response to this question: first of all, the population investigated in this study is high proficient in the L2 (the non-significant correlations between training experience or lextale scores might simply depend on the small sample, and to the little variability within the sample itself) ; second, cognitive resources were not tested/modulated in this study, thus it is unclear if, or how, they contributed to the effect.

Response: Thank you for your comments. First of all, in order to address your concerns, and those of the other reviewer, we expanded our sample size and included a clear speech baseline. We now include 36 participants in the original experiment (listening vs. consecutive interpreting in noisy speech) and we had another group of 36 participants perform the same tasks in clear speech. We recruited participants from the Faculty of Multilingual Communication, Translation and Interpreting, as well as from the participant pool of the Psychology Faculty at the University of Ghent and we did not prescreen participants, except including in our recruitment information that participants needed a good command of English in order to be able to listen to sentences in English and repeat them in Dutch. 

We have now carried out an analysis of the LexTale scores and predictive fixations across both groups (Noisy and Clear speech), thus including 72 participants. We found a correlation between the extent of predictive fixations and the LexTale results. When we considered the noisy and clear speech groups separately, we saw that this correlation was present in the noisy, but not in the clear speech group (lines 687 - 700 and Figure 10). We consider that this could be because in noisy speech, L2 proficiency may be linked to the extent of prediction during comprehension (lines 820-821).

We have also clarified our conclusions. We explain that L2 speakers make predictions in noise, showing that they engage in top-down processing in noisy speech. We have moderated our conclusion to say that our findings show that either if cognitive resources are being used to process noisy speech, L2 listeners still have sufficient resources for prediction, or alternatively that semantic prediction does not require cognitive resources (lines 741 - 743).

Reviewer: So, while there is evidence for prediction, it is not clear if this is due to high performance in the L2 or to the cognitive resources involved in the process, or to other factors. And I do not think that anything can be concluded with respect to this issue.

Response: Our study provides novel evidence of L2 prediction in noise. We believe that this is of interest because it shows that L2 listeners engage in top-down processing even in challenging conditions. The study thus contributes to a body of literature illustrating that prediction regularly takes place during language comprehension. 

It is true that there are still some unanswered questions about whether prediction requires cognitive resources and what level of L2 proficiency is required for prediction, and we suggest in our conclusions that, now that L2 prediction in noise has been established, its limits could be tested by e.g., assessing the minimum level of L2 proficiency necessary for prediction and the range of SNRs within which prediction in L2 is detected.

Prior to running this study, however, it was not clear whether L2 listeners predict in noise, and there were both theoretical and practical reasons for our choice of (relatively) proficient L2 speakers. 

In theoretical terms, we wanted to be sure that participants did comprehend the sentences. Otherwise, any lack of predictive eye movements could simply have been due to lack of comprehension of the sentences. We have clarified this in the text (lines 329-332).

In practical terms, we also had to recruit a group who could carry out the interpreting part of the task (so again, comprehending the sentences was necessary) (lines 329-332). 

However, we also do not believe that our L2 speakers differ from other groups of adult L2 speakers commonly recruited for psycholinguistic studies. We base this assumption on the fact that participants were not pre-screened for proficiency and yet systematically scored at least 70% on the LexTale test (bar two participants).

Given that our correlation analysis revealed a link between the extent of prediction and L2 proficiency (as measured by LexTale score), future studies could indeed consider whether even low proficiency L2 speakers predict in noise, or what the minimum L2 proficiency required to predict is. 

Reviewer: Furthermore, I am not convinced that the type of task implemented in this study requires additional cognitive resources (as assumed by the authors): if it is true that it is challenging, due to the introduced noise, it might simply disrupt the standard process for lexical access (via phonological cues), thus demanding for more top-down processes (based on semantic cues) to repair the noise (which in turn might be linked to language proficiency, not cognitive resources).

Response: We hope that the argument presented in the text is now clearer. We now show clearly how previous literature has found that L2 speakers do not make use of context during listening in noise as L1 listeners do. So, while it has been shown that L1 listeners make use of top-down processes to "repair" the incoming speech, the same is not true of L2 speakers. Hence, it was not clear whether L2 listeners would make predictions in noise (see sub-section "Top-down language processing in noise in L2", starting line 161). We did find a correlation between proficiency and extent of semantic prediction in noise, so L2 proficiency and extent of prediction in noise may indeed be linked (see lines 687 onwards). However, a different type of study would be required to tease apart a potential effect of L2 proficiency from a potential effect of cognitive resources on predictive processing in noise in L2. As you say, prediction could be linked directly to L2 proficiency, or it could be that proficient L2 listeners are able to devote more resources to prediction, because comprehension is less costly.

Reviewer: In some passages, the authors seem to assume that prediction is costly, in contrast with the standard assumption that prediction speeds up processing (and it has been also shown for children). Please clarify.

Response: We do not see a dichotomy here. Prediction may be costly, but it may also speed up processing. For example, the literature shows that people predict less when under cognitive load (e.g., Ito, Pickering & Corley, 2017) or that literate adults predict more than illiterate adults (Mishra, Singh, Pandey & Huettig, 2012), or that higher working memory individuals make stronger predictions (Huettig & Janse, 2016). On the other hand, reading studies show that people read predictable words more quickly (e.g., Rayner, Slattery, Drieghe & Liversedge, 2011). We could for example suppose that prediction requires working resources (ahead of hearing a predictable word) but that it then speeds up processing when people encounter the predictable word. 

Reviewer: In this respect, I would be more cautious in the conclusions, for example when the authors state “It also demonstrates that semantic prediction takes place even when cognitive resources are limited, and leaves open the possibility that phonological prediction may require cognitive resources.”

Response:Thank you. We have adopted a more cautious wording in our conclusions, stating lines 741 to 743 that "This [] suggests either that if cognitive resources are being used to process noisy speech, L2 listeners still have sufficient resources for prediction, or alternatively that semantic prediction does not require cognitive resources. Re. the question of phonological prediction, after testing a group of L2 listeners in clear speech, we also did not find any evidence of phonological prediction. It may therefore be that phonological prediction does not take place regularly in L2 comprehension - regardless of noise.

Reviewer: As for the null results reported: although in general I do favor reporting null results, I am skeptical that these are reliable in this case, due to the little sample tested and the many manipulations introduced/conditions tested. Although the authors argue for the reliability of their participants’ sample size (in line with previous reported studies) I believe that testing less than 30 participants with eye-tracking measures do not provide enough statistical power to gain reliable results, especially in the case post hoc analyses are conducted on a limited number of datapoints, as it is the case here.

Response:In line with your recommendations and those of the other reviewer, we increased our sample size and included a group who listened only in clear speech. The original (noisy speech) experiment now includes 36 participants, and the clear speech group also includes 36 participants. Our sample size is thus now significantly larger than that of previous studies which have found significant results (see lines 316-319).

Reviewer: Furthermore, I wonder whether the null result obtained in comparing the two tasks might be due to the background of the participants: being trained as interpreters, they might approach the listening task in an “interpreter” mode, and this might have obscured a potential difference.

Response: Some participants were students of interpreting who were tested during the first semester of their training, and had between 1 and 4 months training (average of around 2 months). As such, it was unlikely that they would approach the task differently from the other participants. However, we carried out a correlation analysis to investigate this, and found no link between the amount of training participants had received in consecutive interpreting and their extent of semantic prediction.

Reviewer: In general, I wonder why the factors tested in the additional analyses were not introduced as predictors or random factors in the main models. I think this would make the data treatment clearer and more robust.

We have changed the treatment of the data in line with this recommendation, and also in light of your comment (below) that it would be better to focus on the main questions asked. 

Response: For the main analyses, we now included task order as an independent variable. We check for an interaction, and as we find none, we then run the simpler analysis which looks at whether fixation proportions depend on condition. We therefore remove the analysis where we considered the effect of task order on its own.

We also removed the analysis of the timing of the start of interpretation, as it did not correspond to one of our main questions. 

However, we do not include all factors in our main analyses as our main analyses are set up to investigate specific research questions - namely "Do L2 speakers make predictive fixations in noisy conditions? Do predictive fixations depend on task?" Participants' L2 proficiency and training (if any) in interpreting was not linked to our main hypotheses.

This participant information was initially collected to provide additional background information on the group tested. However, since we did not find any difference between the consecutive and listening tasks, or between the noisy/clear speech studies, we then considered whether there was any evidence to suggest an effect of training on predictive fixations in the consecutive task. Since we found no phonological competitor effect, we considered whether this might have been linked to L2 proficiency. We now clarify the reasoning for these analyses better (lines 688 to 691 and 706 to 708).

Reviewer: Also, I think that a control group of L1 English speakers and/or more variability within the L2 speakers (with respect to their proficiency in the L2) is needed to get a baseline and answer some of the questions addressed in the paper.

Response: We have now created a baseline by including a no-noise condition. We now include more L2 speakers in each group and as such there is more variability. As we explained above, the main point is that L2 listeners make semantic predictions in noise - future studies could consider how L2 proficiency affects prediction, but this was not the object of our paper. 

Reviewer: Overall, I find the paper quite difficult to follow, due to the many questions addressed at the beginning (most of which remain, in fact, without a robust answer), and the different main and additional analyses presented, most of which did not return significant results.

I suggest either focusing only on the main question addressed (do (proficient) L2 speakers rely on prediction during sentence processing in noisy (more challenging) conditions?) or expanding the groups so as to increase statistical power, also including L1 speakers and/or L2 speakers with more variable linguistic (and training) background.

Response: We have now added an L2 group in a quiet condition, which makes it possible to consider whether the addition of noise changes predictive behaviour. We have also removed the exploratory analysis which considered whether the onset time of the consecutive interpreting was linked to prediction - because it is not part of our main question, and added task order as an independent variable to the first analysis, rather than carrying out a separate analysis for this. 

Both groups are significantly larger than the group in our original submission (36 in each group as compared to 24 in the original submission). There is more variability, because the L2 speakers include students from the MA in Translation, Interpreting and Communication as well as participants recruited through the Psychology Faculty's participant pool and as such have a variable training background. The linguistic background remains similar, because we have used a group of L2 Dutch-English University students.

We have also streamlined the arguments in our paper (thereby removing 1000 words from the manuscript), added additional sub-section titles to make these arguments easier to follow, revised "The Current Study" section to state our research questions and hypotheses more clearly and revised our conclusions. We hope that these changes to the write-up make the paper easier to follow.

Reviewer: Other points:

- I suggest changing the example in the paper, since this particular example contains a flow in the stimuli: it is not only a “mouth”, it is an “open mouth”, thus the cue is double in this case (dentist primes mouth; open primes an open (mouth)). As far as I can tell from the list of stimuli in the appendix this doesn’t apply to every example (although in other cases as well there is a clear double (or even triple) cue pointing to the target: e.g., 10. honey + stung  bee; 18. Student +library +read book

Response:In fact, the predictability is normed across all sentences and this is what is important for our study, rather than why the predictable words are predictable. The mean predictability is 84.3% (SD: 17.21%). 

In this specific example, the mouth is open because this is the only way to have an image that people consistently identify as being a mouth (a closed mouth would likely be identified by many people as depicting "lips"). 100% of respondents in our norming study identified this image as "mouth" and not "open mouth".

However, we also appreciate your point, and with the aim of making the example as clear and representative of our stimuli set as possible, we now use the example "Bob proposed and gave her a ring that had cost him half his monthly wage". This sentence had a cloze probability of 85% which corresponds to mean predictability in the study.

Reviewer: What about cross-linguistic competitors (in Dutch?). Nothing is said about this, was this controlled for in selection of the stimuli? (for example, both the Dutch translations of mouth and mouse begin with an “m”; bed and ladder in English are very similar to their Dutch translations, but the phonological English competitors are not)—given that Dutch is specifically involved in one of the tasks, these similarities might introduce an additional (not controlled?) confound

Response: We understand why you ask this question, because it is true that both languages are activated in this bilingual setting. However, this was not controlled for as we were only interested in whether Dutch speakers pre-activate the form of a word when listening in their L2. 

As you point out, in some cases, the Dutch and English phonology of the predictable word may have aligned, and in some cases not. However, any phonological effect would still have meant that participants pre-activate the form of a word before hearing it (based on their comprehension of the English sentence). 

In the event, despite any possible reinforcing effect of phonology being shared between Dutch and English, we did not find a convincing competitor effect.

Reviewer: Given the very small set of stimuli presented, I wonder whether seeing the same array of objects in the visual display (albeit in different positions) had an impact on the participants’ performance. We know form other studies that even young children DO remember what has been shown to them before! This is not standard practice in visual world experiment - to control for visual salience typically a Latin square design is employed and the same array of objects is shown in different conditions between, not within subjects. Please clarify this choice.

Response: In fact, our study does employ a Latin-Square design, and we would argue that the set of stimuli is not particularly small - during the experiment participants complete 60 trials and view 120 different images (30 different visual scenes).

We agree that in some other, well-known visual world studies (e.g., Altmann and Kamide, 1999) the manipulation is different. For instance, in Altmann and Kamide's seminal visual word prediction study, the audio stimuli is manipulated within participants to manipulate condition. Half hear the constraining word "eat" and the other half hearing the non-constraining word "move" - and the visual stimuli remains the same, with all participants viewing the scene with the boy, the cake and the other objects. 

Our study is also a Latin Square. Instead of manipulating the audio stimuli, we manipulate the visual stimuli (within participants) to manipulate condition. All participants hear the same sentences, and then one third of participants see the Target object, one third see the Competitor object, and one third see the Unrelated object. 

This design has two main advantages. First, it allows us to ensure that fixations on a competitor object are not swamped by looks to a target object. For instance, Dahan and Tanenhaus (2004) found that participants only fixated a predictable target word in a constraining condition (and not a competitor word) when both were present on the screen. The other advantage is that when two interest areas are both present on the same screen, and fixations between them are compared, the data points for the different conditions are dependent, because when one interest area is fixated, the other necessarily is not (see Ito and Knoeferle, 2022). In our design, the interest areas compared are independent.

It is true that the main disadvantage of this design is that there might be a bias to fixate on one object more than another because participants find it more visually appealing, rather than because they predict a given word. We control for such visual bias by showing participants the same scene twice, once with a predictable and once with a neutral sentence. This design has previously been used by e.g., Ito et al. (2018), Rommers et al. (2013). Sentences are counterbalanced so that visual stimuli appear either as a filler first or as an experimental item first. Thus, there is a counterbalancing to ensure that participants do not always see the target item before it appears in the predictable sentence. Our analysis shows that participants do not generally fixate on target or English competitor objects when they appear with a neutral sentence. 

Reviewer #2: This paper presents evidence that people predict upcoming words when listening to their second language in noisy conditions. In an eye-tracking experiment using the visual world paradigm, participants looked to target images, corresponding to upcoming English words, prior to the onset of those words in high-cloze sentences. The participants were high-proficiency L2 English speakers, and the experiment manipulated whether they listened for comprehension or interpreted those sentences into their L1 (Dutch), manipulated within subjects in two blocks. This consecutive interpreting task did not affect fixation on target images, nor were participants more likely to look to phonological competitors compared to unrelated competitors. However, participants who fixated on target images earlier were slower to begin interpreting, and participants who performing the interpreting task in the first block (i.e., listened only for comprehension in the second half of the experiment) fixated on phonological competitor images more than participants who interpreted in the second block did.

This paper replicates previous findings that people predict upcoming words even in their L2, and it builds on that finding by demonstrating that such prediction occurs even in noise. Given the null effect of interpretation on fixation (and the null effect of fixation on phonological competitors vs unrelated competitors), the key question seems to be whether adding noise during listening is a substantial novel contribution. Having listened to some of the audio files kindly made available on OSF, I’m skeptical; the experience is much like listening to someone speaking quietly. Without a manipulation of the noise condition (e.g., energetic vs informational masking, or varying the SNR), it is hard to contrast these results with L2 listening in previous studies, especially given that the participants in this experiment were high-proficiency speakers of a closely related language (unlike Ito et al., 2018, and Zhao et al., 2022).

Response: Thank you for these comments. We have now additionally tested a group of 36 participants in a no-noise condition to provide a baseline. This allowed us to check whether a group of participants with a similar language profile/background predicted differently in a no-noise condition. 

We must however disagree that demonstrating L2 prediction in noise is not novel. To the best of our knowledge, this is the first paper that shows L2 prediction in noisy conditions.

There are various justifications for the SNR chosen (see lines 418 to 427). First, we wanted to choose an SNR at which participants could still hear the stimuli, but which would make them more challenging to listen to (we did not wish to block comprehension). In addition, we wanted the noise level to relate to conditions that people might experience while interpreting consecutively. If interpreters cannot hear their interlocutor well enough, then they will not be able to provide an interpretation. The same is true in fact of this experiment - if participants had not been able to understand most of the sentence, they would not have been able to provide an interpretation. In fact, we still had to exclude one participant because they did not provide an interpretation 50% of the time, and the participant said this was because they could not hear the stimuli. 

Having established that prediction in noise can take place in L2, our study now opens the door to future research of the type you suggest. For instance, is there an SNR at which participants comprehend speech but at which prediction is limited? Do L2 listeners still predict in noise when their L1 and L2 are not closely related? And is there a minimum level of L2 proficiency necessary for prediction?

Reviewer: The sample size (N = 24) is also concerning, especially because the experiment was not pregregistered. The authors based their sample size on Ito et al. (2018), but that study recruited a total of 48 participants (albeit two groups of 24, only one of which comprised L2 speakers, with 16 items). This experiment added another (within-subject) manipulation and included 30 items, for (I believe) a total of 120 observations per condition. It is almost certainly underpowered. I appreciate the difficulty of recruiting from a specialized population (L1 Dutch interpreters), but the exploratory analyses require replication before they can be accepted with any confidence. The headline effect of fixation on target images is more robust, but again, neither L2 status nor amount/type of noise was manipulated in the design.

Response: Thank you. We agree that increasing the number of observations per condition provides a more reliable result - particularly in case of the null results. Thus we increased the number of participants in our first study to 36 for a total of 180 observations per condition (almost double the number of observations in Ito et al. 2018, who had 96 observations per condition in each study). We also introduced a clear-speech baseline study, for which we tested a further 36 participants.

Reviewer: I am also curious about the presentation of the visual stimulus at 1s prior to the onset of the target word. To my knowledge, this is different from common VWP practice where the visual stimulus is presented prior to the onset of the sentence. I would like to author to justify the divergence of their setup from the common practice. Given the sentences, it is likely that the visual stimulus was presented AFTER the sentence started, which may discourage people from looking too much at the pictures (which may then drown any phonological effect they were looking for). Also, I think it is also important to describe when the pictures were presented relative to the onset of the sentence, given that the target word might occur in different positions in the sentence.

Response: The longer the preview period, the more opportunity participants have to form expectations about linguistic content based on visual information (rather than linguistic information). While it is not possible to exclude the effect that the visual context has on linguistic processing, we attempted to reduce its effect by using a short preview time of 1000ms. We explain this in lines 455 to 457. In addition, we showed a visual array (four unrelated pictures) rather than a visual scene (as in e.g., Altmann & Kamide, 1999, where the boy was shown sitting in the middle of several toys). This is an additional measure to reduce the effect of the visual context on predictive eye movements (see lines 371 to 373)

The information about timing of onset of the predictable word in the sentences is presented in the Stimuli section (lines 357 to 359). Mean onset of the predictable word was 6.26 seconds (SD: 2.00, Range 3.04 to 11.3). All of the visual stimuli were therefore presented after the sentence started. 

With regard to the phonological effect, Ito et al. (2018) found such an effect in L1 speakers using the same design (with 1000ms preview time). If it is the case that an effect in L2 listeners is only present when stimuli are presented prior to the onset of the sentence, it might be that phonological prediction in L2 only takes place when there is a supportive visual context. Another experiment could test this.

Reviewer: I recommend that the authors conduct a follow-up experiment to increase confidence in the exploratory findings and/or to better demonstrate the impact of noise on L2 prediction. As it stands, the novelty of adding energetic masking during L2 listening is questionable, and the sample size is too small to conclude that consecutive interpreting does not affect prediction or that the exploratory effects will replicate.

Response: Thank you. We have followed up on our experiment in line with your suggestions. We have now sampled two groups of 36 participants in both noisy and clear conditions. We hope that this increases confidence in our results, and better demonstrates that, while L2 listeners predict in noise, there is no evidence that noise influences L2 prediction at this SNR, that a consecutive interpreting task affects prediction and that L2 speakers make phonological predictions.

---

## [Decision Letter · Decision Letter 1]

10 Jul 2023

Purposeful listening in challenging conditions: a study of prediction during consecutive interpreting in noise

PONE-D-22-18894R1

Dear Dr. Amos,

We’re pleased to inform you that your manuscript has been judged scientifically suitable for publication and will be formally accepted for publication once it meets all outstanding technical requirements.

In amending your manuscript, please also consider the very minor comments by Reviewer 1.

Kind regards,

Simone Sulpizio

Academic Editor

PLOS ONE

Additional Editor Comments (optional):

Reviewers' comments:

Reviewer's Responses to Questions

**Comments to the Author**

1. If the authors have adequately addressed your comments raised in a previous round of review and you feel that this manuscript is now acceptable for publication, you may indicate that here to bypass the “Comments to the Author” section, enter your conflict of interest statement in the “Confidential to Editor” section, and submit your "Accept" recommendation.

Reviewer #1: All comments have been addressed

Reviewer #2: (No Response)

2. Is the manuscript technically sound, and do the data support the conclusions?

Reviewer #1: Yes

Reviewer #2: Yes

3. Has the statistical analysis been performed appropriately and rigorously? 

Reviewer #1: Yes

Reviewer #2: Yes

4. Have the authors made all data underlying the findings in their manuscript fully available?

Reviewer #1: Yes

Reviewer #2: Yes

5. Is the manuscript presented in an intelligible fashion and written in standard English?

Reviewer #1: Yes

Reviewer #2: Yes

6. Review Comments to the Author

Reviewer #1: I am satisfied with the authors' responses to my previous comments and with the revised version of the manuscript.

I think the paper is now more focused and methodologically sound (with the addition of more participants and the baseline no noise condition).

Minor comments:

- I would change L1 to monolinguals on line 58

- I would add a short summary of the structure of the paper after line 88 (there are many sections/subsections; since these are not numbered it is still difficult to grasp on the overall structure of the manuscript)

- add "in" on line 832 ("even in communicative settings")

- spell out SNRs on line 836

Reviewer #2: The authors have successfully resolved my comments and I am happy to see the publication of the research.

7. PLOS authors have the option to publish the peer review history of their article (what does this mean?). If published, this will include your full peer review and any attached files.

Reviewer #1: No

Reviewer #2: **Yes: **Zhenguang Cai

---

## [Editor Report · Acceptance letter]

12 Jul 2023

PONE-D-22-18894R1 

Purposeful listening in challenging conditions: a study of prediction during consecutive interpreting in noise 

Dear Dr. Amos:

I'm pleased to inform you that your manuscript has been deemed suitable for publication in PLOS ONE. Congratulations! Your manuscript is now with our production department. 

Kind regards, 

on behalf of

Dr. Simone Sulpizio 

Academic Editor

PLOS ONE